# Pharmaceutical enterprises integrity supervision strategy when considering rent-seeking behavior and government reward and punishment mechanism

Yanhua Chen[1], Lilong Zhu[1,2]*

1 School of Business, Shandong Normal University, Ji'nan, China, 2 Quality Research Center, Shandong Normal University, Ji'nan, China

* zhulilong2008@126.com, zhulilong@sdnu.edu.cn

## Abstract

The integrity of pharmaceutical enterprises is crucial to public health, social stability, and national security, consistently garnering attention from both the government and society. The efficiency of pharmaceutical integrity supervision is closely linked to government oversight mechanisms and the behaviors of third-party testing agencies. This study constructs an evolutionary game model that incorporates rent-seeking dynamics and introduces a reward-punishment mechanism, integrating drug production enterprises, third-party testing agencies, government regulators, and drug wholesale enterprises. By solving for the stable equilibrium points of each participant's strategic choices and analyzing the stability of strategy combinations using Lyapunov's first method, the study employs Matlab 2022b for simulation analysis to verify the impact of various decision variables on the strategic choices of different entities. The findings reveal that: 1) The rejection of rent-seeking by third-party testing agencies enhances the incentives for drug production enterprises to operate with integrity, indicating that the government should increase penalties for accepting rent-seeking behavior. 2) Drug wholesale enterprises' reporting likelihood increases production enterprises' integrity and third-party testing agencies' rejection of rent-seeking, inversely tied to reporting costs. 3) Reducing the costs of stringent government supervision and increasing the speculative costs of rent-seeking for third-party testing agencies help prevent dishonest practices among drug production enterprises. 4) A well-designed reward and punishment mechanism facilitates a synergistic environment of government supervision, self-discipline among pharmaceutical enterprises, and social harmony. This paper enriches the theoretical foundation of pharmaceutical integrity supervision and offers pertinent countermeasures and recommendations.

## 1 Introduction

The integrity of pharmaceutical enterprises is an embodiment of national governance capabilities and an important guarantee for the healthy development of the economy and society. The lack of integrity in the pharmaceutical field severely disrupts the normal functioning of

**Data availability statement:** All the data is available and within the manuscript.

**Funding:** This work was supported by the National Social Science Fund of China under grant No.20BGL272, Natural Science Foundation of Shandong Provincial in China under grant No.ZR2024MG048, and University Youth Science and Technology Innovation Team Project of Shandong Province in China under grant No.2021RW010.

**Competing interests:** The authors have declared that no competing interests exist.

the medical industry, particularly when rent-seeking behaviors come into play. Rent-seeking behavior refers to the acquisition of economic benefits by enterprises or individuals through non-market means (e.g., bribery or interest transfer) instead of achieving profit growth through efficiency improvements or innovation. In the pharmaceutical industry, drug production enterprises can influence the test results of third-party testing agencies through rent-seeking behavior, which not only distorts normal medical practices and diminishes the quality of medical services but also allows unqualified drugs to enter the market, posing a serious threat to patient safety. From an ethical perspective, rent-seeking behavior places economic interests above patients' health and is a betrayal of professional spirit and social responsibility. At the same time, rent-seeking behavior leads to unfair allocation of medical resources, exacerbates excessive medical treatment and drug price increases, increases the economic burden on patients, and ultimately hinders the healthy and sustainable development of the pharmaceutical industry. Against this background, it is urgent to strengthen corporate governance to ensure that corporate behaviors comply with ethical norms and legal regulations, thereby reshaping industry trust and promoting a virtuous cycle in the pharmaceutical industry.

Government departments have actively implemented measures to enhance the integrity supervision of pharmaceutical enterprises. In April 2021, the World Health Organization (WHO) officially released the "Good Quality Management Practices for Drug Supervision" to improve the ability to supervise drugs during and after the event (https://www.who.int/). In order to ensure patient medication safety, in February 2022, the U.S. Food and Drug Administration (FDA) officially released the final guidance on "Patient-Centered Drug Research and Development: Identifying Methods that Matter to Patients" (https://www.fda.gov/). In March 2020, the European Medicines Agency (EMA) released "EMA Regulatory Science 2025: Strategic Reflections," aiming to establish a more adaptable regulatory system and encourage drug innovation (https://www.ema.europa.eu/en/homepage). In order to promote scientific integrity supervision of pharmaceutical enterprises, in February 2023, China emphasized improving a new regulatory mechanism that takes "double randomization, one openness" supervision and "Internet + supervision" as the basic means, supplements key supervision, and is based on integrity supervision (https://www.gov.cn/).

The reward and punishment mechanism serves as an effective tool for enhancing the efficiency of integrity supervision. Therefore, this article considers the rent-seeking behavior of third-party testing agencies, introduces a government reward and punishment mechanism, and constructs a four-party evolutionary game model involving drug production enterprises, third-party testing agencies, government regulators and drug wholesale enterprises. The stable equilibrium points of the strategic choices of each game subject are solved, and the stability of the strategy combination and the influence of each key element on the strategy evolution are analyzed, aiming to solve the following three problems. First, how does the reward and punishment mechanism function in order to avoid dishonest management by drug production enterprises? Second, how does the participation of government regulators and third-party testing agencies affect the strategic choices of drug production enterprises? Third, how does the reporting behavior of drug wholesale enterprises affect the strategic choices of drug production enterprises and third-party testing agencies?

The remainder of this article is organized as follows. The second part combs and reviews the relevant literature; the third part puts forward hypotheses and constructs a four-party evolutionary game model involving drug production enterprises, third-party testing agencies, government regulators and drug wholesale enterprises; the fourth part analyzes the stability of the strategic choices of the four participating subjects; the fifth part solves the stable equilibrium point of the strategic choices of each game subject according to Lyapunov's first rule, and analyzes the stability of the strategy combination; the sixth part uses Matlab 2022b to conduct

simulation analysis of the constructed game model; the seventh part discusses and puts forward relevant suggestions; the last part provides conclusions.

## 2 Literature review

### 2.1 Integrity issues of pharmaceutical enterprises

The integrity issues of pharmaceutical enterprises involve complex interest relationships and multiple subjects. The untrustworthy behaviors of enterprises directly impact the quality and safety of drugs, thereby further endangering public health. For instance, untrustworthy enterprises may utilize inferior or unqualified raw materials to cut down production costs, thereby directly threatening the lives and health of patients (Xu Y., et al, 2023; Rong J., et al, 2020) [1,2]. Furthermore, the lack of integrity in pharmaceutical enterprises may lead to a decline in social trust, exacerbate the unfairness of pharmaceutical resources, and ultimately harm social stability and national security (Helmut B., et al, 2024) [3]. However, restricted by production costs, enterprise scale and high investment in drug research and development, opportunistic behaviors and rent - seeking phenomena in enterprises occur frequently (Busby J.S., 2019) [4]. Existing research highlights that unqualified or counterfeit drugs constitute a significant proportion of supply chains in many countries, undermining market fairness and consumer rights (Baptiste et al., 2022) [5]. In particular, the COVID - 19 pandemic in 2019 exposed a large number of untrustworthy problems, making the government pay more attention to the integrity issues of pharmaceutical enterprises (Ranjana P., et al, 2023; Bao Y., et al, 2020) [6,7].

### 2.2 Government integrity supervision

Although government regulation is regarded as an effective way to ensure the honest operation of pharmaceutical enterprises (Magdalena B., et al, 2021) [8], cases of regulatory failures, such as the vaccine fraud incident of Changchun Changsheng Bio - tech Co., Ltd. in China in 2018, rent-seeking incidents in Pfizer and Johnson & Johnson, have exposed the blind spots of government regulation (Flynn M., 2018) [9]. These incidents highlight the issues of information transparency and enforcement in government regulation and emphasize the importance of improving the regulatory system. The Suppression of dishonest behavior by pharmaceutical enterprises requires a complete regulatory system(Jing L et al, 2021) [10]. Therefore, the government should combine punishment mechanisms with incentive mechanisms for appropriate supervision(Zhang S and Zhu L, 2021) [11], in order to pursue the dynamic balance of integrity supervision, and introduce new drug production models by strengthening management methods(Godman B et al, 2021; Khan A et al, 2024; Nunavath S R et al, 2024) [12–14], reduce drug costs(Ana S et al, 2023) [15], and further improve the multi-agent collaborative supervision model(Zhang M et al, 2019) [16], reasonable use of new media means(Khoo K Y et al, 2024) [17], improve the public's ability to identify substandard drugs(Zhang S and Zhu L, 2022) [18], and leverage multiple subjects to play a role together(Faris D E and S. MKF, 2021) [19]. In addition, regulatory measures such as improving the transparency and accountability of pharmaceutical enterprises (Paschke A et al, 2018) [20] and implementing digital transformation strategies(Hole G et al, 2021) [21] can also effectively reduce the probability of pharmaceutical enterprises losing trust.

### 2.3 Participation of third-party testing agencies

As neutral evaluation agencies, third-party testing agencies can effectively mitigate the impact of dishonest behaviors of pharmaceutical enterprises (Bujar M et al, 2020) [22], ensure the fairness and authority of supervision, and promote the improvement of government supervision efficiency(Parker L et al, 2020) [23]. However, the acceptance of rent-seeking by

third-party testing agencies will cause substandard drugs to flow into the market, bring risks to patients' health and safety, and hinder the healthy development of the pharmaceutical industry (Wang ZY and Cao Y, 2023) [24]. Strengthening the supervision of rent-seeking behavior, increasing penalties(Jain N et al, 2021; Newton P N et al, 2020) [25,26], guiding the public to actively participate, exposing and reporting non-compliance(Bhaskar V et al, 2019) [27] are conducive to promoting trust building in the pharmaceutical industry.

The research by Zhang and Zhu [11] has provided valuable insights into the co-regulation of pharmaceutical enterprises. However, their study has certain limitations. Firstly, their research only considered the relationships among local governments, drug enterprises, and third-party testing agencies, while ignoring the important role of drug wholesale enterprises in the supervision of drug quality. Secondly, their model assumptions are relatively simple and do not fully take into account the whistleblowing behavior of drug wholesale enterprises and its impact on the strategic choices of all parties. Compared with their study, this research introduces drug wholesale enterprises as a new participant and constructs a more complex four-party evolutionary game model, which can more comprehensively reflect the actual game relationship in the pharmaceutical supply chain.

All in all, the existing literature has made great research contributions to promoting the integrity operation of pharmaceutical enterprises, but there are still deficiencies. Firstly, in the research on the integrity issues of pharmaceutical enterprises [1–7], the existing literature seldom discusses from the perspective of multi-agent collaborative supervision, and the research on multi-agent behavior games and strategy selections is relatively blank. Secondly, the research on government integrity supervision is very rich [8–21], but it mostly focuses on the design and optimization of the supervision mechanism, while the dynamic effect analysis of the reward and punishment mechanism is still insufficient. Finally, although the existing research has emphasized the supervisory role of third-party agencies [22–27], there are still gaps in the analysis of how they make strategy selections under the reward and punishment mechanism and how they interact with the whistle - blowing behavior of drug wholesale enterprises.

The research contributions of this paper are as follows: Firstly, considering multi-agent collaborative supervision, an evolutionary game model participated by drug production enterprises, third-party testing agencies, government regulators and drug wholesale enterprises is constructed. Secondly, considering the influence of the behaviors of third-party testing agencies and drug wholesale enterprises on the strategy selections of other participating agents under the reward and punishment mechanism, the stable equilibrium points of strategy choices under different conditions are solved. Finally, the influence of each decision variable is analyzed, and Matlab 2022b is used for simulation analysis to verify the effectiveness of the model, and countermeasures and suggestions for the integrity supervision of pharmaceutical enterprises are put forward.

## 3  Model hypotheses and construction

### 3.1  Problem description

In the context of globalization, the pharmaceutical industry is facing increasingly complex regulatory challenges. In particular, rent-seeking behavior erodes the integrity supervision system, threatening public health and market fairness. How to promote the integrity of pharmaceutical enterprises and build a fair and transparent market environment through effective reward and punishment mechanisms has become a key issue that urgently needs to be addressed. In response to this real-world scenario, this paper constructs an evolutionary game model involving drug production enterprises, third-party testing agencies, government regulators and drug wholesale enterprises. Through model analysis, we aim to gain a deeper

understanding of the strategy choice and evolution process of each participating party under the reward and punishment mechanism, and explore how to optimize the reward and punishment mechanism to improve the efficiency of integrity supervision of pharmaceutical enterprises. This paper will provide policy makers with decision-making bases based on real market behaviors, guiding the government, enterprises and third-party agencies on how to formulate and implement more effective reward and punishment policies in specific situations, in order to promote integrity operation, maintain market order, safeguard public health, and build a harmonious pharmaceutical market environment. Therefore, the construction and analysis of this model will directly benefit from in-depth exploration of specific real-world scenarios, providing policy makers with clearer and more practical guiding directions.

Evolutionary game theory, as a powerful analytical tool, has the core advantage of being able to depict and predict the dynamic strategic interactions among multiple agents. It is especially suitable for research scenarios involving complex decision-making environments such as integrity supervision of pharmaceutical enterprises. By using evolutionary game theory, we can deeply analyze the decision-making process of each participating party when facing the reward and punishment mechanism and reveal the mechanism of action of the reward and punishment mechanism on strategy evolution. Therefore, evolutionary game theory has strong applicability and effectiveness in solving the problem of integrity supervision of pharmaceutical enterprises. The logical relationship between the four parties in the evolutionary game of integrity supervision of pharmaceutical enterprises is shown in Fig 1.

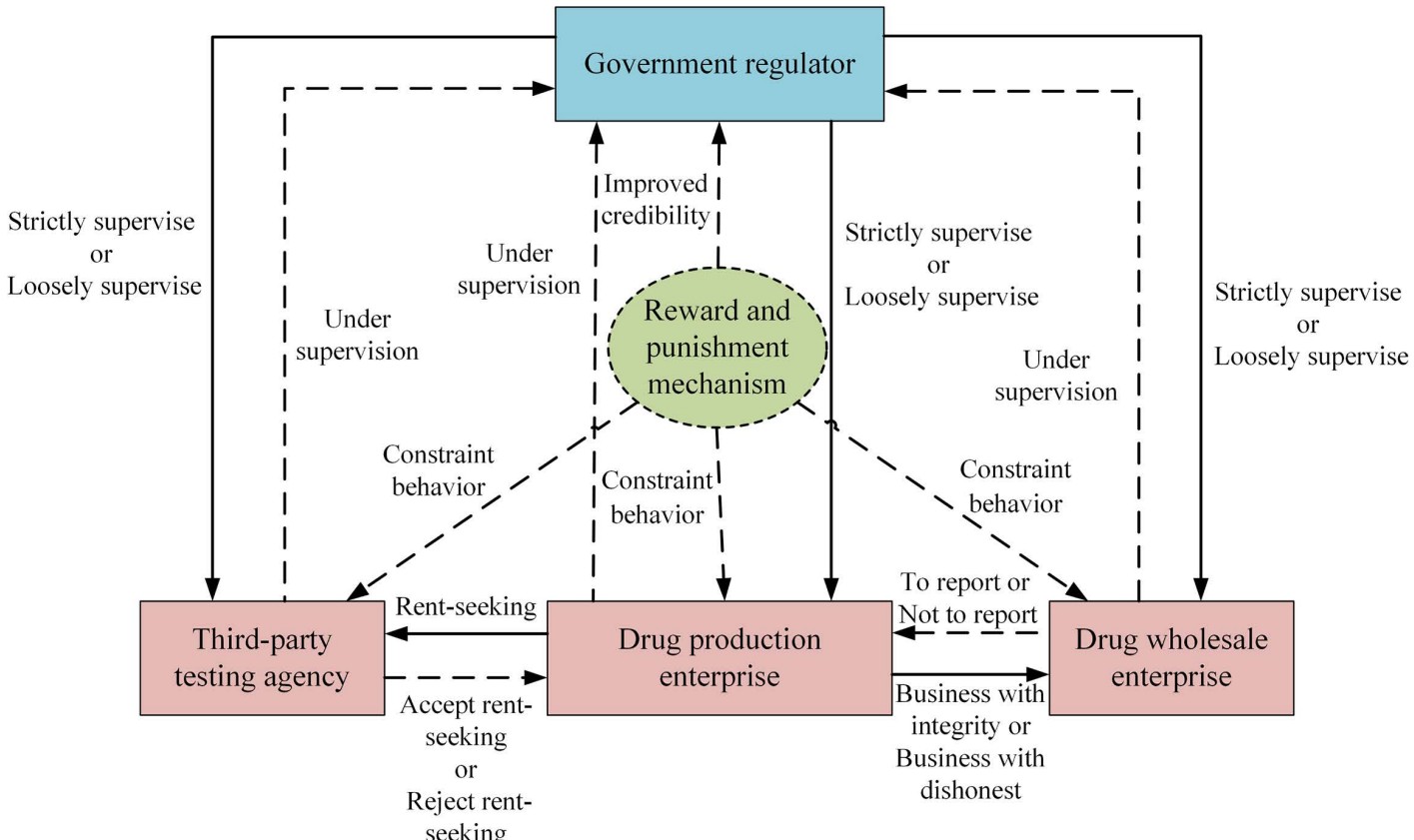

**Fig 1. Four-party game relationship.** Fig 1 is a structural diagram of the relationship between drug production enterprises, third-party testing agencies, government regulators and drug wholesale enterprises that introduce government reward and punishment mechanisms under rent-seeking behavior.

## 3.2 Model hypotheses

Considering the secretive nature of rent-seeking behavior and the sensitivity of relevant data, this study extensively referenced officially released data when setting the model parameters, such as "2022 Annual Pharmaceutical Supervision and Administration Statistics Data", "2023 Department Budget of the National Medical Products Administration", "FDA Annual Performance and Financial Report", "Overview of Health Economic Data in the UK" and "World Health Statistics", to ensure the rationality of key parameters. Additionally, based on a literature review and comprehensive analysis of the industry situation, supplemented by field research and interviews with industry experts, we have formed a preliminary estimate of the model parameters and made the following hypotheses.

H$1$ This article has four participating subjects: drug production enterprises, third-party testing agencies, government regulators and drug wholesale enterprises. The strategic choice of the government regulator is {strict supervision, loose supervision}. Assume that the probability of the government regulator choosing strict supervision is $g(0 \leq g \leq 1)$ and the probability of choosing loose supervision is $1-g$. The strategic choice of a drug production enterprise is {integrity management, dishonest management}. Integrity management refers to the production of high-quality drugs, and dishonest management refers to the production of low-quality drugs. Assume that the probability of a drug production enterprise choosing honest management is $x(0 \leq x \leq 1)$, and the probability of dishonest management is $1-x$. The strategy choice of the third-party testing agency is {accept rent-seeking, reject rent-seeking}. Accepting rent-seeking means that the third-party testing agency passes the testing of low-quality drugs to obtain marketing authorization. Assume that the probability of the third-party testing agency choosing to accept rent-seeking is $y(0 \leq y \leq 1)$, the probability of rejecting rent seeking is $1-y$. The strategic choice of the drug wholesale enterprise is {reporting, not reporting}, assuming that the probability of the drug wholesale enterprise choosing to report is $z(0 \leq z \leq 1)$, and the probability of not reporting is $1-z$.

H$2$ The wholesale price of drugs is denoted as $W$. The production cost for low-quality drugs is $C_l$, while the cost for high-quality drugs is $C_h$, where $C_h > C_l$. When drug production enterprises operate dishonestly, they will seek rent from third-party testing agencies to pass the test. At this time, the rent-seeking cost is $B_t$, and the speculation cost of falsifying production records, false publicity, etc. is $C_p$, where $B_t + C_p < C_h - C_l$.

H$3$ The testing income of the third-party testing agency is $V_t$, and the testing cost is $C_t$. When a drug production enterprise operates dishonestly, if a third-party testing agency accepts rent-seeking, it will need to forge test records and issue false test reports. Assume that the speculation cost of accepting rent-seeking is $C_s$. If third-party testing agencies refuse to seek rent and government regulators strictly supervise, low-quality drugs cannot be marketed.

H$4$ Drug wholesale enterprises sell drugs to hospitals and drug retailers, and their profits are $R$. When a drug wholesale enterprise wholesales high-quality drugs, its operating conditions are good, and its reputation value premium is $I$. When a drug wholesale enterprise wholesales low-quality drugs, it discovers drug quality problems through daily sampling, feedback from hospitals and pharmaceutical retail enterprises, etc. If the drug wholesale enterprise chooses to report, the reporting cost is $C_r$. If a drug wholesale enterprise chooses not to report it, it will sell substandard drugs, resulting in a reputational loss of $L$.

H$5$ The cost of strict supervision by government regulators is $C_1$, and the cost of loose supervision is $C_2(C_2 < C_1)$. When government regulators strictly supervise, drug production enterprises will be fined $F_p$ for dishonest managements and required to compensate drug wholesale enterprise $R_l$; third-party testing agencies will be fined $F_t$ for accepting rent-seeking; drug wholesale enterprises will be rewarded $M_w$ for reporting, failure to report will result in a fine $F_w$. When the government regulators loose supervision and do not take any reward or punishment

measures, if a drug wholesale enterprise wholesales low-quality drugs and chooses to report it, the higher-level government will hold the government regulators accountable for their dereliction of duty (interviews, admonitions, removal from office, etc.), the amount of punishment is $F_g$.

*H6* The flow of high-quality drugs into the market will help protect people's lives and health, maintain good market order, and increase the credibility of government regulators by $S$. The flow of low-quality drugs into the market harms patients' rights and disrupts market order, and the credibility of government regulators will decrease by $D_g$. Relevant parameters and descriptions are shown in Table 1.

### 3.3 Model construction

Based on the above hypotheses, this paper constructs a four-party mixed strategy game matrix of drug production enterprises, third-party testing agencies, government regulators and drug wholesale enterprises, as shown in Table 2.

## 4 Analysis of the strategic choice stability

### 4.1 Drug production enterprises' strategic choice stability

The replication dynamic equation and the first derivative of the strategic choices of drug production enterprises are:

$$F(x) = dx/dt = x(E_x - \overline{E}) = x(1-x)(E_x - E_{1-x})$$
$$= x(1-x)[F_p(z - gz + g) - W(yg - g) + C_l + C_p - C_h + yB_t + ygR_l + z(1-g)R_l] \quad (1)$$

$$F'(x) = (1-2x)[F_p(z - gz + g) - W(yg - g) + C_l + C_p - C_h + yB_t + ygR_l + z(1-g)R_l] \quad (2)$$

According to the stability theorem of differential equations, when a drug production enterprise chooses the "integrity management" strategy and is in a stable state, the probability must satisfy: $F(x) = 0$ and $F'(x) < 0$.

**Proposition 1** When $z > z_0$, the stable strategy of drug production enterprises is "integrity management"; when $z < z_0$, the stable strategy of drug production enterprises is "dishonest management"; when $z = z_0$, the stable strategy cannot be determined; where the threshold is

**Table 1. Description of relevant parameters.**

| symbol | Describe | symbol | describe |
|---|---|---|---|
| $g$ | The probability that the government will choose strict regulation | $F_g$ | The amount of fines imposed by superior governments on government regulators |
| $x$ | The probability of drug production enterprises choosing to operate with integrity | $C_t$ | Speculation costs of drug production enterprises |
| $y$ | The probability that a third-party testing agency chooses to accept rent seeking | $F_w$ | Fines for drug wholesale enterprises |
| $z$ | The probability of drug wholesale enterprises choosing to report | $R$ | Profit of drug wholesale enterprises |
| $C_l$ | The cost of producing low-quality drugs | $F_t$ | Fines for third-party testing agencies |
| $C_h$ | The cost of producing high-quality drugs | $M_w$ | Incentive amount for drug wholesale enterprises |
| $B_t$ | Rent-seeking cost | $D_g$ | Decline in government credibility |
| $V_t$ | Testing income | $C_1$ | The cost of strict government regulation |
| $C_s$ | The speculative costs of accepting rent-seeking | $C_2$ | The cost of loose government regulation |
| $W$ | Drug wholesale prices | $L$ | Reputational loss |
| $C_r$ | Testing cost | $F_p$ | Fines for drug production enterprises |
| $I$ | Reputation value premium | $S$ | Increase government credibility |
| $R_l$ | Amount of compensation | $C_p$ | Reporting cost |

**Table 2. Game matrix of four parties.**

| Choice of strategy | Third-party testing agencies | Government regulators | | | |
|---|---|---|---|---|---|
| | | Strictly supervise $g$ | | Loosely supervise $1-g$ | |
| | | Drug wholesale enterprises report $z$ | Drug wholesale enterprises do not report $1-z$ | Drug wholesale enterprises report $z$ | Drug wholesale enterprises do not report $1-z$ |
| Drug production enterprises integrity management $x$ | Accept rent-seeking $y$ | $W-C_h$ <br> $V_t-C_t$ <br> $S-C_1$ <br> $R+I$ | $W-C_h$ <br> $V_t-C_t$ <br> $S-C_1$ <br> $R+I$ | $W-C_h$ <br> $V_t-C_t$ <br> $S-C_2$ <br> $R+I$ | $W-C_h$ <br> $V_t-C_t$ <br> $S-C_2$ <br> $R+I$ |
| | Refuse rent-seeking $1-y$ | $W-C_h$ <br> $V_t-C_t$ <br> $S-C_1$ <br> $R+I$ | $W-C_h$ <br> $V_t-C_t$ <br> $S-C_1$ <br> $R+I$ | $W-C_h$ <br> $V_t-C_t$ <br> $S-C_2$ <br> $R+I$ | $W-C_h$ <br> $V_t-C_t$ <br> $S-C_2$ <br> $R+I$ |
| Drug production enterprises dishonest management $1-x$ | Accept rent-seeking $y$ | $W-C_l-B_t-C_p-F_p-R_l$ <br> $V_t-C_t-C_s+B_t-F_t$ <br> $F_p+F_t-C_1-D_g-M_w$ <br> $R-C_r+M_w-L+R_l$ | $W-C_l-B_t-C_p-F_p-R_l$ <br> $V_t-C_t-C_s+B_t-F_t$ <br> $F_p+F_t-C_1-D_g+F_w$ <br> $R-L+R_l-F_w$ | $W-C_l-B_t-C_p-F_p-R_l$ <br> $V_t-C_t-C_s+B_t-F_t$ <br> $F_p+F_t-C_2-D_g-F_g-M_w$  $R-C_r+M_w-L+R_l$ | $W-C_l-B_t-C_p$ <br> $V_t-C_t-C_s+B_t$ <br> $-C_2-D_g$ <br> $R-L$ |
| | Refuse rent-seeking $1-y$ | $-C_l-C_p-F_p$ <br> $V_t-C_t$ <br> $-C_1+F_p$ <br> $0$ | $-C_l-C_p-F_p$ <br> $V_t-C_t$ <br> $-C_1+F_p$ <br> $0$ | $W-C_l-C_p-F_p-R_l$ <br> $V_t-C_t$ <br> $F_p-C_2-D_g-F_g-M_w$ <br> $R-C_r+M_w-L+R_l$ | $W-C_l-C_p$ <br> $V_t-C_t$ <br> $-C_2-D_g$ <br> $R-L$ |

$$z_0 = \frac{gF_p - W(yg-g) + C_l + C_p - C_h + yB_t + ygR_l}{(g-1)R_l + F_p(g-1)}.$$

*Proof* Let $H(z) = F_p(z - gz + g) - W(yg-g) + C_l + C_p - C_h + yB_t + ygR_l + z(1-g)R_l$, since $\partial H(z)/\partial z > 0$, then $H(z)$ is an increasing function with respect to $z$. Therefore, when $z < z_0$, $H(z) < 0$, $F(x)|_{x=0} = 0$ and $F'(x)|_{x=0} < 0$, then $x = 0$ has stability; when $z > z_0$, $H(z) > 0$, $F(x)|_{x=1} = 0$ and $F'(x)|_{x=1} < 0$, then $x = 1$ has stability; when $z = z_0$, $H(z) = 0$, $F(x) = 0$ and $F'(x) = 0$, at this time, drug production enterprises cannot determine a stable strategy. Certification completed.

Proposition 1 shows that an increase in the probability of reporting by drug wholesale enterprises will change the stability strategy of drug production enterprises from dishonest management to integrity management, and vice versa. Therefore, the non-reporting behavior of drug wholesale enterprises is not conducive to the honest operation of pharmaceutical enterprises. Government regulators should strictly supervise pharmaceutical enterprises by strengthening the awareness of rights protection of drug wholesale enterprises, broadening the rights protection channels of drug wholesale enterprises and strengthening law enforcement.

In summary, the response function of $x$ is:

$$x = \begin{cases} 0 & if \quad z < \dfrac{gF_p - W(yg-g) + C_l + C_p - C_h + yB_t + ygR_l}{(g-1)R_l + F_p(g-1)} \\[2ex] (0,1) & if \quad z = \dfrac{gF_p - W(yg-g) + C_l + C_p - C_h + yB_t + ygR_l}{(g-1)R_l + F_p(g-1)} \\[2ex] 1 & if \quad z > \dfrac{gF_p - W(yg-g) + C_l + C_p - C_h + yB_t + ygR_l}{(g-1)R_l + F_p(g-1)} \end{cases} \tag{3}$$

According to Proposition 1, the phase diagram of drug production enterprises strategic choice is shown in Fig 2.

It can be seen from Fig 2 that the volume of part $V_{A1}$ is the probability that the drug production enterprise chooses the "integrity management" strategy, and the volume of $V_{A0}$ is the probability that the drug production enterprise chooses the "dishonest management" strategy. Moreover,

$$
\begin{aligned}
V_{A1} &= \int_0^1 \int_{\frac{C_l+C_p-C_h+yB_t}{-R_l-F_p}}^1 \frac{gF_p - W(yg-g) + C_l + C_p - C_h + yB_t + ygR_l}{(g-1)R_l + F_p(g-1)} dzdx \\
&= \frac{gF_p - W(yg-g) + C_l + C_p - C_h + yB_t + ygR_l}{(g-1)R_l + F_p(g-1)}\left(1 - \frac{C_l+C_p-C_h+yB_t}{-R_l-F_p}\right)
\end{aligned}
\tag{4}
$$

$$
\begin{aligned}
V_{A0} &= 1 - V_{A1} \\
&= 1 - \frac{gF_p - W(yg-g) + C_l + C_p - C_h + yB_t + ygR_l}{(g-1)R_l + F_p(g-1)}\left(1 - \frac{C_l+C_p-C_h+yB_t}{-R_l-F_p}\right)
\end{aligned}
\tag{5}
$$

**Corollary 1.1** When the fine amount of a drug production enterprise for dishonest management is $F_p > F_{p0}$, the drug production enterprise will choose the "integrity management" strategy; when $F_p < F_{p0}$, the drug production enterprise will choose the "dishonest management" strategy. Among them, the threshold is

$$
F_{p0} = \frac{C_l + C_p - C_h - W(yg-g) + yB_t + ygR_l + z(1-g)R_l}{gz - g - z}.
$$

*Proof* According to Proposition 1, when $H(z) = 0$, we can get $F_{p0} = \frac{C_l + C_p - C_h - W(yg-g) + yB_t + ygR_l + z(1-g)R_l}{gz - g - z}$, and $\partial H(z)/\partial F_p > 0$, so $H(z)$ is an increasing function of $F_p$. When $F_p > F_{p0}$, $H(z) > 0$, at this time $F(x)|_{x=1} = 0$ and $F'(x)|_{x=1} < 0$; when $F_p < F_{p0}$, $H(z) < 0$, at this time $F(x)|_{x=0} = 0$ and $F'(x)|_{x=0} < 0$. Certification completed.

Corollary 1.1 shows that when the fine amount $F_p$ imposed by the government regulator on a drug production enterprise for dishonest operation is greater than the threshold $F_{p0}$, it can ensure that the drug production enterprise chooses the "integrity management"

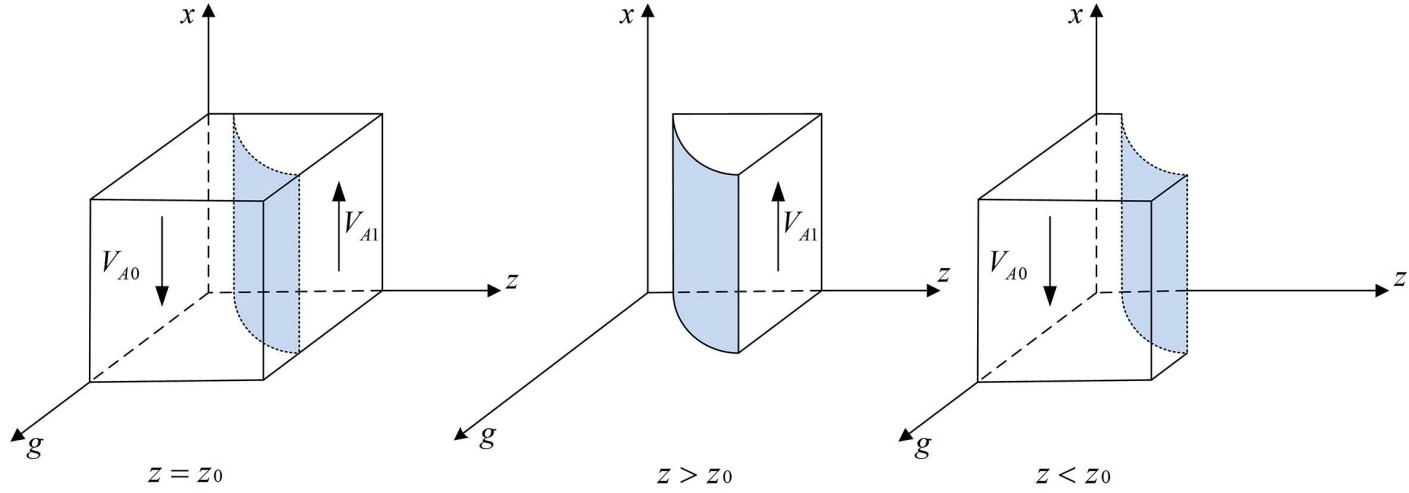

**Fig 2. Phase diagram of drug production enterprises strategic choices.** Fig 2 is the phase diagram that shows the evolutionary trend of drug production enterprises 's strategy obtained by calculating the response function of the probability of drug production enterprises choosing the "integrity management" strategy.

strategy. When $F_p$ is less than the threshold $F_{p0}$, the drug production enterprise will choose the "dishonest management" strategy. Therefore, when drug production enterprises operate dishonestly, government regulators should increase punishments to urge drug production enterprises to operate with integrity, maintain good market order, and protect the lives and health of the people.

**Corollary 1.2** When the compensation amount of the drug production enterprise to the drug wholesale enterprise is $R_l > R_{l0}$, the drug production enterprise will choose the "integrity management" strategy; when $R_l < R_{l0}$, drug production enterprises will choose the "dishonest management" strategy. Among them, the threshold is

$$R_{l0} = \frac{F_P(z - gz + g) - W(yg - g) + C_l + C_p - C_h + yB_t}{-yg - z + gz}.$$

*Proof* According to Proposition 1, when $H(z) = 0$, we can get
$R_{l0} = \frac{F_P(z - gz + g) - W(yg - g) + C_l + C_p - C_h + yB_t}{-yg - z + gz}$, and $\partial H(z) / \partial F_P > 0$, so $H(z)$
is an increasing function of $R_l$. When $R_l > R_{l0}$, $H(z) > 0$, at this time $F(x)|_{x=1} = 0$ and

$F'(x)|_{x=1} < 0$; when $R_l < R_{l0}$, $H(z) < 0$, at this time $F(x)|_{x=0} = 0$ and $F'(x)|_{x=0} < 0$. Certification completed.

Corollary 1.2 shows that when the compensation amount $R_l$ of the drug production enterprise to the drug wholesale enterprise is greater than the threshold $R_{l0}$, it can ensure that the drug production enterprise chooses the "integrity management" strategy; when $R_l$ is less than the threshold $R_{l0}$, the drug production enterprise will choose the "dishonest management" strategy. Therefore, when drug production enterprises operate dishonestly, government regulators should increase the amount of compensation that the drug production enterprises pay to drug wholesale enterprises, strengthen their moral awareness, and make them consciously protect the rights and interests of patients.

## 4.2 Third-party testing agencies' strategic choice stability

The replication dynamic equation and first derivative of third-party testing **agencies'** strategic choices are:

$$\begin{aligned} F(y) = dy / dt &= y(E_y - \overline{E}) = y(1 - y)(E_y - E_{1-y}) \\ &= y(1 - y)(1 - x)(-zF_t - gF_t - C_s + B_t + gzF_t) \end{aligned} \tag{6}$$

$$F'(y) = (1 - 2y)(1 - x)(-zF_t - gF_t - C_s + B_t + gzF_t) \tag{7}$$

According to the stability theorem of differential equations, the probability that third-party testing agencies chooses the "accept rent-seeking" strategy to be in a stable state must satisfy: $F(y) = 0$ and $F'(y) < 0$.

**Proposition 2** The thresholds are $z_1 = \frac{B_t - gF_t - C_s}{F_t - gF_t}$, $g_0 = \frac{B_t - zF_t - C_s}{F_t - zF_t}$. When $g < g_0$, $z < z_1$, the stability strategy of the third-party testing agency is "accept rent-seeking". When $g > g_0$, $z > z_1$, the stability strategy of the third-party testing agency is "reject rent-seeking". When $g = g_0$, $z = z_1$, the stability strategy cannot be determined.

*Proof*

Let $G(g,z) = (1 - x)(-zF_t - gF_t - C_s + B_t + gzF_t)$, since $\partial G(z) / \partial z < 0$, $\partial G(g) / \partial g < 0$, then $G(g,z)$ is a decreasing function with respect to $g$, $z$. Therefore, when $g > g_0$, $z > z_1$, $G(g,z) < 0$, $F(y)|_{y=0} = 0$ and $F'(y)|_{y=0} < 0$, then $y = 0$ has stability; when $g < g_0$, $z < z_1$,

$G(g,z)>0$, $F(y)|_{y=1}=0$ and $F'(y)|_{y=1}<0$, then $y=1$ has stability; when $g=g_0$, $z=z_1$, $G(g,z)=0$, then $F(y)=0$ and $F'(y)=0$, at this time the third-party testing agency cannot determine the stability strategy. Certification completed.

Proposition 2 shows that an increase in the probability of reporting by drug wholesale enterprises will change the stability strategy of third-party testing agencies from accepting rent-seeking to rejecting rent-seeking. In the same way, the increased probability of strict supervision by government regulators will change the stability strategy of third-party testing agencies from accepting rent-seeking to rejecting rent-seeking. Therefore, government regulators should actively choose a "strict supervision" strategy and encourage third-party testing agencies to standardize their testing behavior by strengthening the awareness of rights protection of drug wholesale enterprises and rewarding their reporting behaviors.

In summary, the response function of $y$ is:

$$y=\begin{cases} 0 & if \quad g>\dfrac{B_t-zF_t-C_s}{F_t-zF_t} \\[2mm] (0,1) & if \quad g=\dfrac{B_t-zF_t-C_s}{F_t-zF_t} \\[2mm] 1 & if \quad g<\dfrac{B_t-zF_t-C_s}{F_t-zF_t} \end{cases} \tag{8}$$

According to Proposition 2, the phase diagram of third-party testing agencies strategic choice is shown in Fig 3.

It can be seen from Fig 3 that the volume of part $V_{B1}$ is the probability that the third-party testing agency chooses to accept rent-seeking, and the volume of part $V_{B0}$ is the probability that it chooses to reject rent-seeking. Moreover,

$$V_{B1}=\int_0^1\int_0^{\frac{B_t-C_s}{F_t}}\frac{B_t-zF_t-C_s}{F_t-zF_t}dzdy=\frac{B_t-C_s}{F_t}+\left(1-\frac{B_t-C_s}{F_t}\right)\ln\frac{C_s-B_t+F_t}{F_t} \tag{9}$$

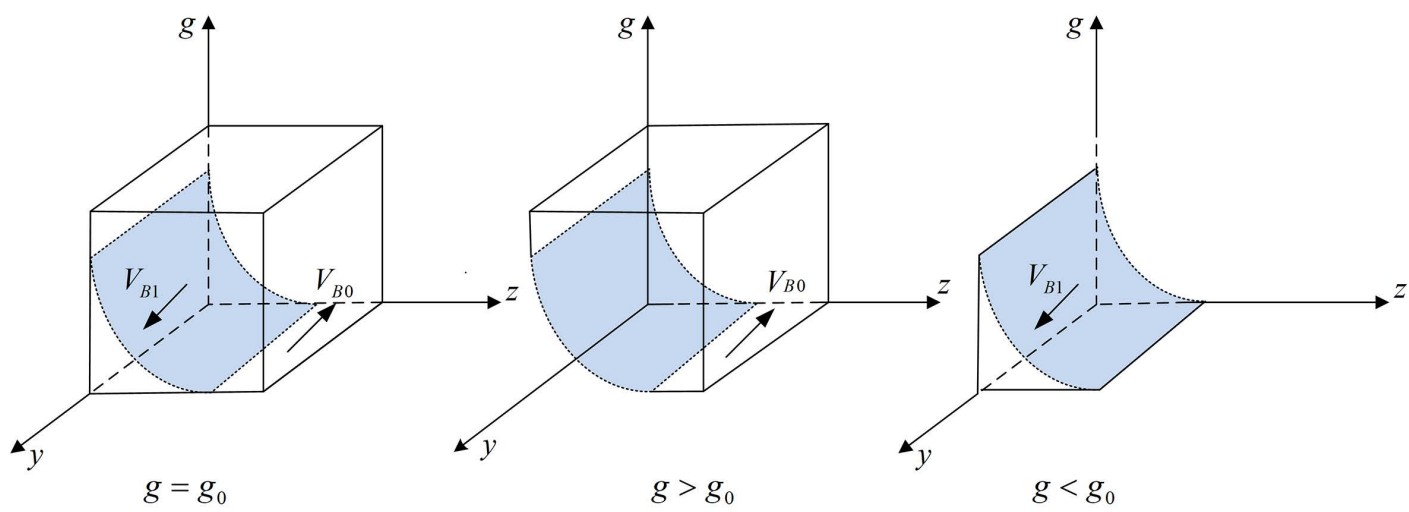

**Fig 3. Phase diagram of third-party testing agencies strategic choices.** Fig 3 is the phase diagram that shows the evolutionary trend of third-party testing agencies' strategy obtained by calculating the response function of the probability of third-party testing agencies choosing the "accept rent-seeking" strategy.

$$V_{B0} = 1 - \frac{B_t - C_s}{F_t} - (1 - \frac{B_t - C_s}{F_t}) \ln \frac{C_s - B_t + F_t}{F_t} \tag{10}$$

**Corollary 2.1** The probability of a third-party testing agency accepting rent-seeking is positively related to its rent-seeking income and inversely proportional to the amount of fines imposed by government regulators for accepting rent-seeking behavior.

*Proof*

The third-party testing agency selects the probability $V_{B1}$ of "accepting rent-seeking", and calculates the first-order partial derivatives of $B_t$ and $F_t$ respectively, and we get:

$$\frac{\partial V_{B1}}{\partial B_t} = -\frac{1}{F_t} \ln \frac{C_s - B_t + F_t}{F_t} > 0 \tag{11}$$

$$\frac{\partial V_{B1}}{\partial F_t} = \frac{B_t - C_s}{F_t^2} \ln \frac{C_s - B_t + F_t}{F_t} - \frac{1}{F_t} < 0 \tag{12}$$

**Corollary 2.1** shows that the greater the rent-seeking income of a third-party testing agency and the smaller the fine amount when accepting rent-seeking, the more willing it is to accept rent-seeking. Therefore, severe penalties will help reduce the probability of third-party testing agencies accepting rent-seeking. Government regulators should increase strict supervision of third-party testing agencies, increase fines for rent-seeking behavior, and promote fair testing by third-party testing agencies.

**Corollary 2.2** The probability of a third-party testing agency accepting rent-seeking is inversely proportional to its speculation cost when accepting rent-seeking.

*Proof*

The third-party testing agency selects the probability $V_{B1}$ of "accepting rent-seeking", and calculates the first-order partial derivative of $C_s$, we get:

$$\frac{\partial V_{B1}}{\partial C_s} = \frac{1}{F_t} \ln \frac{C_s - B_t + F_t}{F_t} < 0 \tag{13}$$

**Corollary 2.2** shows that the higher the speculative cost of a third-party testing agency accepting rent-seeking, the smaller the probability that it will choose to accept rent-seeking. Therefore, increasing the speculative costs of rent-seeking accepted by third-party testing agencies will help reduce their speculative behavior, ensure drug safety, and maintain market order.

## 4.3 Government regulators' strategic choice stability

The replication dynamic equation and first-order derivative of the government regulator's strategic choices are:

$$\begin{aligned} F(g) &= dg / dt = g(E_g - \overline{E}) = g(1-g)(E_g - E_{1-g}) \\ &= g(1-g)[(1-x)(yF_t - yD_g - yzM_w - zF_p + zF_g + zM_w + C_2 + D_g - yzF_t + F_p - C_1) \\ &\quad - x(C_1 - C_2) + (1-x)y(1-z)F_w] \end{aligned} \tag{14}$$

$$\begin{aligned} F'(g) &= (1-2g)[(1-x)(yF_t - yD_g - yzM_w - zF_p + zF_g + zM_w + C_2 + D_g \\ &\quad - yzF_t + F_p - C_1) - x(C_1 - C_2) + (1-x)y(1-z)F_w] \end{aligned} \tag{15}$$

According to the stability theorem of differential equations, the probability that the government regulator chooses the "strict supervision" strategy to be in a stable state must satisfy:
$F(g) = 0$ and $F'(g) < 0$.

**Proposition 3** When $x < x_0$, the stabilization strategy of the government regulator is "strict supervision"; when $x > x_0$, the stabilization strategy of the government regulator is "loose supervision"; when $x = x_0$, the stability strategy cannot be determined; where the threshold is

$$x_0 = \frac{yF_t - yD_g - yzM_w - zF_p + zF_g + zM_w + D_g - yzF_t + F_p + yF_w - yzF_w + C_2 - C_1}{yF_t - yD_g - yzM_w - zF_p + zF_g + zM_w + D_g - yzF_t + F_p + yF_w - yzF_w}$$

*Proof*

Let $M(x) = (1-x)(yF_t - yD_g - yzM_w - zF_p + zF_g + zM_w + C_2 + D_g - yzF_t + F_p - C_1) - x(C_1 - C_2) + (1-x)y(1-z)F_w$, since $\partial M(x)/\partial x < 0$, then $M(x)$ is a decreasing function with respect to $x$. Therefore, when $x > x_0$, $M(x) < 0$, $F(g)|_{g=0} = 0$ and $F'(g)|_{g=0} < 0$, then $g = 0$ has stability; when $x < x_0$, $M(x) > 0$, $F(g)|_{g=1} = 0$ and $F'(g)|_{g=1} < 0$, then $g = 1$ has stability; when $x = x_0$, $M(x) = 0$, then $F(g) = 0$ and $F'(g) = 0$, at this time the third-party testing agency cannot determine the stability strategy. Certification completed.

Proposition 3 shows that an increase in the probability of dishonest management by drug production enterprises will change the stabilization strategy of government regulators from loose supervision to strict supervision, and vice versa. Therefore, when the stable strategy of a drug production enterprise is to operate with integrity, government regulators can observe this behavior and adopt a loose regulatory strategy.

In summary, the response function of $g$ is:

$$g = \begin{cases} 0 & \text{if } x > 1 + \dfrac{C_2 - C_1}{yF_t - yD_g - yzM_w - zF_p + zF_g + zM_w + D_g - yzF_t + F_p + yF_w - yzF_w} \\ (0,1) & \text{if } x = 1 + \dfrac{C_2 - C_1}{yF_t - yD_g - yzM_w - zF_p + zF_g + zM_w + D_g - yzF_t + F_p + yF_w - yzF_w} \\ 1 & \text{if } x < 1 + \dfrac{C_2 - C_1}{yF_t - yD_g - yzM_w - zF_p + zF_g + zM_w + D_g - yzF_t + F_p + yF_w - yzF_w} \end{cases} \quad (16)$$

According to Proposition 3, the phase diagram of government regulators strategic choice is shown in Fig 4.

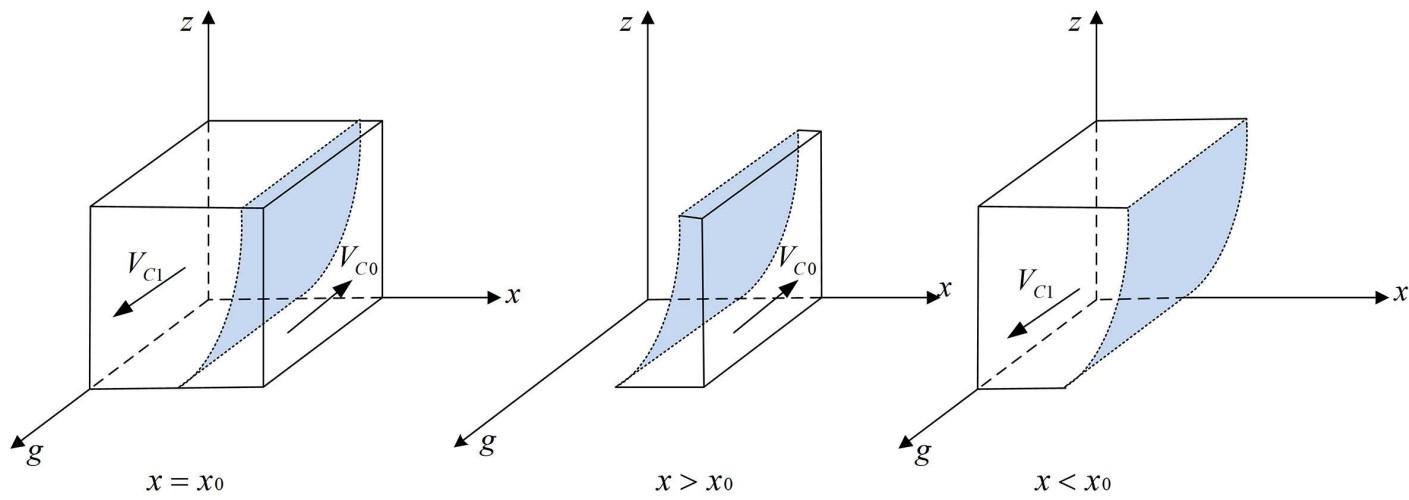

**Fig 4. Phase diagram of government regulators strategic choices.** Fig 4 is the phase diagram that shows the evolutionary trend of government regulators' strategy obtained by calculating the response function of the probability of government regulators choosing the "strict supervision" strategy.

It can be seen from Fig 4 that the volume of part $V_{C1}$ is the probability that the government regulator will choose strict supervision, and the volume of part $V_{C0}$ is the probability of choosing loose supervision. Moreover,

$$V_{C0} = \int_0^1 \int_0^1 1 + \frac{C_2 - C_1}{z(F_g - yM_w - F_p + M_w - yF_t - yF_w) + D_g + F_p + yF_w + yF_t - yD_g} dzdg$$

$$= 1 + \frac{C_2 - C_1}{F_g - yM_w - F_p + M_w - yF_t - yF_w} \ln \frac{F_g - yM_w + M_w + D_g - yD_g}{D_g + F_p + yF_w + yF_t - yD_g} \tag{17}$$

$$V_{C1} = 1 - V_{C0} = \frac{C_1 - C_2}{F_g - yM_w - F_p + M_w - yF_t - yF_w} \ln \frac{F_g - yM_w + M_w + D_g - yD_g}{D_g + F_p + yF_w + yF_t - yD_g} \tag{18}$$

**Corollary 3.1** The probability that government regulators choose "strict supervision" is directly proportional to the amount of fines imposed by the higher-level government for their dereliction of duty.

*Proof*

The probability $V_{C0}$ of the government regulator choosing "loose regulation", and finding the first-order partial derivative of $F_g$, we get:

$$\frac{\partial V_{C0}}{\partial F_g} = \frac{C_1 - C_2}{(F_g - yM_w - F_p + M_w - yF_t - yF_w)^2} \ln \frac{F_g - yM_w + M_w + D_g - yD_g}{D_g + F_p + yF_w + yF_t - yD_g}$$

$$+ \frac{C_2 - C_1}{(F_g - yM_w - F_p + M_w - yF_t - yF_w)(F_g - yM_w + M_w + D_g - yD_g)} < 0 \tag{19}$$

**Corollary 3.1** shows that the greater the fine amount imposed by the superior government on the loose supervision of government regulators, the smaller the probability of loose supervision. Therefore, higher-level governments increasing fines for loose regulatory behaviors will help government regulators strictly supervise the pharmaceutical market, ensure drug safety, and establish good social credibility.

**Corollary 3.2** The probability of government regulators choosing "strict regulation" is negatively related to the additional cost of choosing strict regulation.

*Proof*

The probability $V_{C0}$ of the government regulator choosing "loose regulation", and taking the first-order partial derivative of $C_1 - C_2$, we get:

$$\frac{\partial V_{C0}}{\partial(C_1 - C_2)} = -\frac{1}{F_g - yM_w - F_p + M_w - yF_t - yF_w} \ln \frac{F_g - yM_w + M_w + D_g - yD_g}{D_g + F_p + yF_w + yF_t - yD_g} > 0 \tag{20}$$

**Corollary 3.2** shows that the greater the additional cost for government regulators to choose strict supervision, the greater the probability of choosing loose supervision. Therefore, measures such as simplifying government administrative procedures, improving work efficiency, and increasing the level of digitalization should be used to reduce the cost of strict supervision by government regulators, thereby prompting them to choose a "strict supervision" strategy.

## 4.4 Drug wholesale enterprises' strategic choice stability

The replication dynamic equation and the first derivative of the strategic choices of drug wholesale enterprises are as follows:

$$F(z) = dz / dt = z(E_z - \overline{E}) = z(1 - z)(E_z - E_{1-z})$$

$$= z(1 - z)[(1 - x)(1 - g + gy)(-C_r + R_l + M_w) - (1 - x)ygR_l + (1 - x)ygF_w] \tag{21}$$

$$F'(z) = (1-2z)[(1-x)(1-g+gy)(-C_r+R_l+M_w)-(1-x)ygR_l+(1-x)ygF_w] \quad (22)$$

According to the stability theorem of differential equations, the probability that a drug whole-sale enterprise chooses the "reporting" strategy to be in a stable state must satisfy: $F(z)=0$ and $F'(z)<0$.

**Proposition 4** When $y>y_0$, the stable strategy of drug wholesale enterprises is "reporting"; when $y<y_0$, the stable strategy of drug wholesale enterprises is "not reporting"; when $y=y_0$,

the stable strategy cannot be determined; where the threshold is $y_0 = \dfrac{(g-1)(R_l+M_w-C_r)}{g(F_w+M_w-C_r)}$.

*Proof*

Let $N(y)=(1-x)(1-g+gy)(-C_r+R_l+M_w)-(1-x)ygR_l+(1-x)ygF_w$, since $\partial N(y)/\partial y>0$, then $N(y)$ is a decreasing function with respect to $y$. Therefore, when $y<y_0$, $N(y)<0$, $F(z)|_{z=0}=0$ and $F'(z)|_{z=0}<0$, then $z=0$ has stability; when $y>y_0$, $N(y)>0$, $F(z)|_{z=1}=0$ and $F'(z)|_{z=1}<0$, then $z=1$ has stability; when $y=y_0$, $N(y)=0$, then $F(z)=0$ and $F'(z)=0$, at this time third-party testing agencies cannot determine the stability strategy. Certification completed.

Proposition 4 shows that an increase in the probability that third - party testing agencies reject rent-seeking will change the stable strategy of drug wholesale enterprises from reporting illegal activities to not reporting them, and vice versa. Therefore, the refusal of third-party testing agencies to seek rent is conducive to maintaining the stability of the pharmaceutical market and safeguarding the health of patients.

In summary, the response function of $z$ is:

$$z = \begin{cases} 0 & if \quad y < \dfrac{(g-1)(R_l+M_w-C_r)}{g(F_w+M_w-C_r)} \\[3mm] (0,1) & if \quad y = \dfrac{(g-1)(R_l+M_w-C_r)}{g(F_w+M_w-C_r)} \\[3mm] 1 & if \quad y > \dfrac{(g-1)(R_l+M_w-C_r)}{g(F_w+M_w-C_r)} \end{cases} \quad (23)$$

According to Proposition 4, the phase diagram of drug wholesale enterprises strategic choice is shown in Fig 5.

It can be seen from Fig 5 that the volume of part $V_{D1}$ is the probability of drug wholesale enterprises choosing to report, and the volume of part $V_{D0}$ is the probability of not reporting. Moreover,

$$V_{D1} = \int_0^1 \int_{\frac{R_l+M_w-C_r}{R_l-F_w}}^1 \frac{(g-1)(R_l+M_w-C_r)}{g(F_w+M_w-C_r)} dg dy = \frac{R_l+M_w-C_r}{C_r-F_w-M_w} \ln\frac{R_l-F_w}{R_l+M_w-C_r} + \frac{R_l+M_w-C_r}{F_w-R_l} \quad (24)$$

$$V_{D0} = 1-V_{D1} = 1 - \frac{R_l+M_w-C_r}{C_r-F_w-M_w} \ln\frac{R_l-F_w}{R_l+M_w-C_r} - \frac{R_l+M_w-C_r}{F_w-R_l} \quad (25)$$

**Corollary 4.1** The probability of drug wholesale enterprises choosing to report is inversely proportional to its reporting cost.

*Proof*

The probability $V_{D1}$ of drug wholesale enterprises choosing to report, and finding the first-order partial derivative of $C_r$, we get:

$$\frac{\partial V_{D1}}{\partial C_r} = \frac{F_w-R_l}{(C_r-F_w-M_w)^2} \ln\frac{R_l-F_w}{R_l+M_w-C_r} + \frac{1}{C_r-F_w-M_w} - \frac{1}{F_w-R_l} < 0 \quad (26)$$

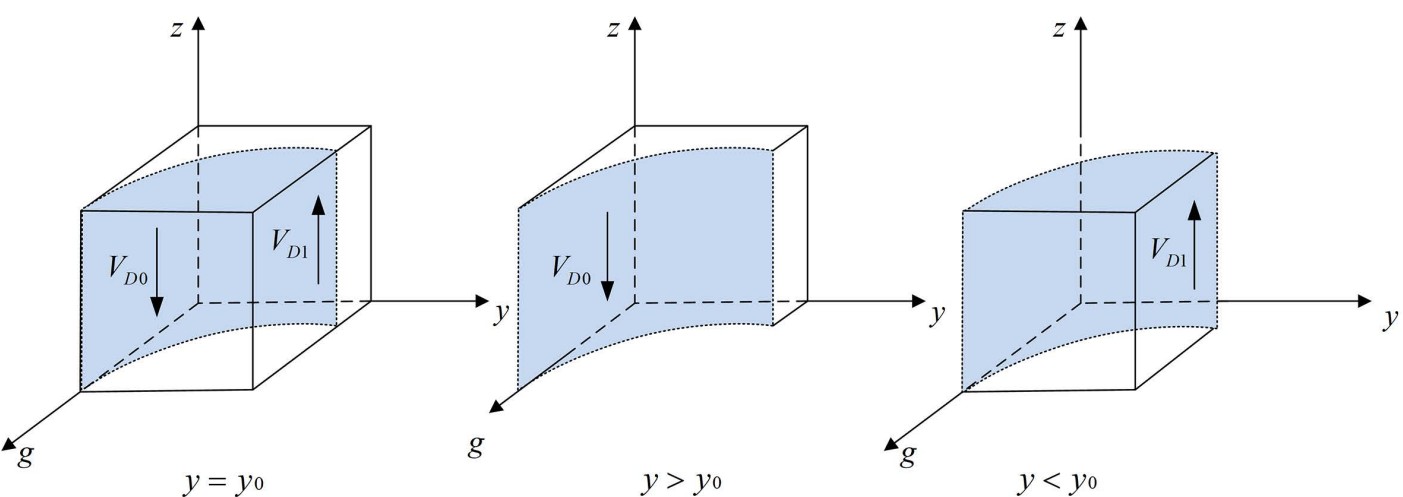

**Fig 5. Phase diagram of drug wholesale enterprises strategic choices.** Fig 5 is the phase diagram that shows the evolutionary trend of drug wholesale enterprises' strategy obtained by calculating the response function of the probability of drug wholesale enterprises choosing the "reporting" strategy.

**Corollary 4.1** shows that the higher the reporting cost of drug wholesale enterprises is, the lower the probability of choosing to report is. Therefore, government regulators should actively take measures to broaden reporting channels for drug wholesale enterprises, improve reporting efficiency, and reduce reporting costs.

**Corollary 4.2** The probability that drug wholesale enterprises choose to report is directly proportional to the amount of reward given by the government regulators for reporting behavior, and directly proportional to the amount of fine imposed by government regulators for non-reporting behavior.

*Proof* The drug wholesale enterprise chooses the probability $V_{D1}$ of "reporting", and calculates the first-order partial derivatives of $M_w$ and $F_w$ respectively, and we get:

$$\frac{\partial V_{D1}}{\partial M_w} = \frac{R_l - F_w}{(C_r - F_w - M_w)^2} \ln \frac{R_l - F_w}{R_l + M_w - C_r} - \frac{1}{C_r - F_w - M_w} + \frac{1}{F_w - R_l} > 0 \tag{27}$$

$$\frac{\partial V_{D1}}{\partial F_w} = \frac{R_l + M_w - C_r}{(C_r - F_w - M_w)^2} \ln \frac{R_l - F_w}{R_l + M_w - C_r} - \frac{R_l + M_w - C_r}{(C_r - F_w - M_w)(R_l - F_w)} - \frac{R_l + M_w - C_r}{(F_w - R_l)^2} > 0 \tag{28}$$

**Corollary 4.2** shows that the higher the reward amount given by the government regulators to drug wholesale enterprises for reporting behavior and the higher the fine amount for non-reporting behavior **is**, the greater the probability that drug wholesale enterprises will choose to report **is**. Therefore, government regulators should set up a reasonable reward and punishment mechanism. Appropriately increasing the intensity of rewards and punishments will help drug wholesale enterprises report drug quality problems, thereby promoting the improvement of integrity level in pharmaceutical enterprises.

## 5 Stability analysis of strategic combination

In the replicated dynamic system of the four-party game between drug production enterprises, third-party testing agencies, government regulators and drug wholesale enterprises, the stability of the strategy combination of the four-party game subjects can be judged according

to Lyapunov's first rule. The stable solution in multi-population evolutionary games is strict Nash equilibrium, and strict Nash equilibrium must be a pure strategy. Therefore, this study will analyze the stability of 16 pure strategy equilibrium points in the four-party evolutionary game.

According to the replication dynamic equations of each game subject, the Jacobian matrix of the replication dynamic system is obtained:

$$J = \begin{vmatrix} \partial F(x)/\partial x & \partial F(x)/\partial y & \partial F(x)/\partial g & \partial F(x)/\partial z \\ \partial F(y)/\partial x & \partial F(y)/\partial y & \partial F(y)/\partial g & \partial F(y)/\partial z \\ \partial F(g)/\partial x & \partial F(g)/\partial y & \partial F(g)/\partial g & \partial F(g)/\partial z \\ \partial F(z)/\partial x & \partial F(z)/\partial y & \partial F(z)/\partial g & \partial F(z)/\partial z \end{vmatrix} \tag{29}$$

## 5.1 Stability of strategic combination in accepting rent-seeking

When the stability strategy of the third-party testing agency is "accepting rent-seeking", i.e., satisfying condition $(1-x)(-zF_t - gF_t - C_s + B_t + gzF_t) > 0$, the asymptotic stability analysis of the equilibrium point of the replicated dynamic system is shown in Table 3.

Condition ①: $B_t + C_l + C_p + F_p + R_l < C_h$, $C_s + F_t < B_t$, $C_1 - C_2 < F_p + F_t + F_w$, $F_w + M_w - C_r < 0$; Condition②: $C_2 + F_p + F_t + F_w < C_1$, $M_w + R_l < C_r$; Condition③: $B_t + C_l + C_p + F_p + R_l < C_h$, $C_s + F_t < B_t$, $F_g < C_1 - C_2$, $M_w + R_l > C_r$; Condition④: $B_t + C_l + C_p + F_p + R_l < C_h$, $C_s + F_t < B_t$, $F_g > C_1 - C_2$, $C_r - M_w < F_w$.

It can be seen from Table 3 that when third-party testing agencies accept rent-seeking, there is no stable strategy combination for drug production enterprises to operate with integrity. It can be seen that when third-party testing agencies accept rent-seeking, the drug market order is chaotic, low-quality drugs flow into the market, and patient safety is difficult to guarantee. At this time, in order to prevent drug production enterprises' dishonest managements from becoming a stable strategy, government regulators should adhere to strict supervision strategies and increase penalties for drug production enterprises' dishonest managements and third-party testing agencies' rent-seeking behavior. And in order to ensure the binding effect of the fine amount, it should at least meet the conditions: $F_p < C_h - R_l - C_p - C_l - B_t$, $F_t > B_t - C_s$ to avoid the equilibrium point $E_1(0,1,1,0)$, $E_3(0,1,0,1)$ or $E_6(0,1,1,1)$ become ESS. In addition, increasing the compensation amount to drug wholesale enterprises, reducing their reporting costs, and increasing rewards for reporting behavior are also strategies to

**Table 3. Asymptotic stability of the equilibrium point in accepting rent-seeking.**

| Equilibrium point | Eigenvalues $\lambda_1, \lambda_2, \lambda_3, \lambda_4$ | Sign | Stability |
|---|---|---|---|
| $E_1(0,1,1,0)$ | $B_t - C_h + C_l + C_p + F_p + R_l$, $C_s - B_t + F_t$, $C_1 - C_2 - F_p - F_t - F_w$, $F_w - C_r + M_w$ | $(\times,\times,\times,\times)$ | It is ESS when ① is satisfied |
| $E_2(0,1,0,0)$ | $B_t - C_h + C_l + C_p$, $C_s - B_t$, $C_2 - C_1 + F_p + F_t + F_w$, $M_w - C_r + R_l$ | $(-,-,\times,\times)$ | It is ESS when ② is satisfied |
| $E_3(0,1,0,1)$ | $B_t - C_h + C_l + C_p + F_p + R_l$, $C_s - B_t + F_t$, $C_2 - C_1 + F_g$, $C_r - M_w - R_l$ | $(\times,\times,\times,\times)$ | It is ESS when ③ is satisfied |
| $E_4(1,1,1,0)$ | $C_h - B_t - C_l - C_p - F_p - R_l$, $0$, $C_1 - C_2$, $0$ | $(\times,0,+,0)$ | Unstable |
| $E_5(1,1,1,1)$ | $C_h - B_t - C_l - C_p - F_p - R_l$, $0$, $C_1 - C_2$, $0$ | $(\times,0,+,0)$ | Unstable |
| $E_6(0,1,1,1)$ | $B_t - C_h + C_l + C_p + F_p + R_l$, $C_s - B_t + F_t$, $C_1 - C_2 - F_g$, $C_r - F_w - M_w$ | $(\times,\times,\times,\times)$ | It is ESS when ④ is satisfied |
| $E_7(1,1,0,1)$ | $C_h - B_t - C_l - C_p - F_p - R_l$, $0$, $C_2 - C_1$, $0$ | $(\times,0,-,0)$ | Unstable |
| $E_8(1,1,0,0)$ | $C_h - B_t - C_l - C_p$, $0$, $C_2 - C_1$, $0$ | $(+,0,-,0)$ | Unstable |

Note: $\times$ indicates that the sign is uncertain; if conditions ①, ②, ③, and ④ are met, they are stable points.

avoid the dishonest management of drug production enterprises, that is, to avoid the equilibrium point $E_2$ (0,1,0,0) becoming ESS, the compensation amount, reporting cost and reward amount must meet the conditions: $M_w + R_l > C_r$ .

The acceptance of rent-seeking by third-party testing agencies has significantly harmed the healthy development of the pharmaceutical industry. Preventing third-party testing agencies from accepting rent-seeking is an important measure to break the chaos in the drug market and improve the integrity of pharmaceutical enterprises. Government regulators should speed up the improvement of the reward and punishment mechanism, and eliminate the occurrence of rent-seeking by third-party testing agencies by increasing the punishment for untrustworthy business behavior and accepting rent-seeking behavior, and increasing the rewards for reporting behavior.

## 5.2 Stability of strategic combination in rejecting rent-seeking

When the stability strategy of the third-party detection agency is "refuse rent-seeking", i.e., the condition is met $(1 - x)(-zF_t - gF_t - C_s + B_t + gzF_t) < 0$ , the asymptotic stability analysis of the equilibrium point of the replicated dynamic system is shown in Table 4.

As can be seen from Table 4, in the case where the third-party testing agency refuses to seek rent, $E_{16}$ (0,0,0,1) may become a stable equilibrium point due to the dereliction of duty by the government regulators. At this time, reducing the cost of strict supervision by government regulators and increasing the penalties for dereliction of duty by government regulators will make the regulatory costs and fines meet the conditions: $C_1 - F_g < C_2 + D_g + M_w$ , which will help to encourage government regulators to strictly supervise the business activities of pharmaceutical enterprises, thus prompting drug production enterprises to choose integrity management strategies.

The dereliction of duty by government regulators will cause low-quality drugs to flow into the market, disrupt the order of the drug market, reduce the integrity of pharmaceutical enterprises, and cause great harm to people's lives and health and the healthy development of the pharmaceutical industry. Measures such as reducing the cost of strict supervision by government regulators and increasing penalties for dereliction of duty can eliminate the occurrence of loose supervision by government regulators and are of great significance to improving the integrity level of the pharmaceutical industry.

**Table 4. Asymptotic stability of the equilibrium point in refusing rent-seeking.**

| Equilibrium point | Eigenvalues $\lambda_1, \lambda_2, \lambda_3, \lambda_4$ | Sign | Stability |
|---|---|---|---|
| $E_9$(1,0,1,0) | $C_h - C_l - C_p - F_p - W$ , 0 , $C_1 - C_2$ , 0 | $(\times, 0, +, 0)$ | Unstable |
| $E_{10}$(0,0,1,0) | $C_l - C_h + C_p + F_p + W$ , $B_t - C_s - F_t$ , $C_1 - C_2 - D_g - F_p$ , 0 | $(+, \times, \times, 0)$ | Unstable |
| $E_{11}$(0,0,1,1) | $C_l - C_h + C_p + F_p + W$ , $B_t - C_s - F_t$ , $C_1 - C_2 - D_g - F_g - M_w$ , 0 | $(+, \times, \times, 0)$ | Unstable |
| $E_{12}$(1,0,0,0) | $C_h - C_l - C_p$ , 0 , $C_2 - C_1$ , 0 | $(+, 0, -, 0)$ | Unstable |
| $E_{13}$(1,0,0,1) | $C_h - C_l - C_p - F_p - R_l$ , 0 , $C_2 - C_1$ , 0 | $(\times, 0, -, 0)$ | Unstable |
| $E_{14}$(0,0,0,0) | $C_l - C_h + C_p$ , $B_t - C_s$ , $C_2 - C_1 + D_g + F_p$ , $M_w - C_r + R_l$ | $(-, +, \times, \times)$ | Unstable |
| $E_{15}$(1,0,1,1) | $C_h - C_l - C_p - F_p - W$ , 0 , $C_1 - C_2$ , 0 | $(-, 0, +, 0)$ | Unstable |
| $E_{16}$(0,0,0,1) | $C_l - C_h + C_p + F_p + R_l$ , $B_t - C_s - F_t$ , $C_2 - C_1 + D_g + F_g + M_w$ , $C_r - M_w - R_l$ | $(\times, \times, \times, \times)$ | It is ESS when ⑤ is satisfied |

Note: $\times$ indicates that the sign is uncertain; if condition ⑤ is met, they are stable points.

Condition ⑤: $C_l + C_p + F_p + R_l < C_h$ , $C_s + F_t > B_t$ , $D_g + F_g + M_w < C_1 - C_2$ , $C_r < R_l + M_w$ .

## 6 Simulation analysis

According to the "2022 Drug Supervision and Administration Statistical Annual Data", regulatory agencies at all levels investigated and dealt with 514 cases of counterfeit and substandard drugs in 2022, with a total value of 430 million CNY. Based on this, this article sets the cost of drug production enterprises producing low-quality drugs for $C_l = 4.3$. According to the "National Medical Products Administration Department Budget for 2023", the market supervision and administration drug affairs budget at the beginning of the year is 450 million CNY. It is assumed that the cost of strict government supervision is $C_1 = 4.5$. Based on relevant literature and combined with the realistic background, other parameters are assigned the following values: $C_h = 19.3$, $W = 25$, $F_p = 4$, $C_p = 2$, $C_2 = 1.5$, $F_t = 4$, $F_g = 5$, $B_t = 5$, $D_g = 3$, $R_l = 2$, $M_w = 3$, $C_r = 6$, $C_s = 1.5$, $F_w = 4$.

### 6.1 The impact of government regulators on the amount of fines and incentives imposed on drug wholesale enterprises

Set $F_w = \{0.0, 3.0, 7.0\}$, $M_w = \{0.0, 3.0, 7.0\}$, the evolution process and results of the main strategies of the four-party game are shown in Fig 6.

It can be seen from Fig 6 that the reward and punishment mechanism of government regulators for drug wholesale enterprises has a positive effect on the integrity management of pharmaceutical enterprises. When the amount of fines and rewards increases, the stability strategy of drug wholesale enterprises changes from not reporting to reporting. At this time, the fluctuation range of the probability that drug production enterprises operate with integrity and the probability of strict government supervision decreases, and the probability of third-party testing agencies rejecting rent-seeking increases. Therefore, government regulators should severely punish drug wholesale enterprises for non-reporting behavior and set up reasonable reward mechanisms to encourage drug wholesale enterprises to report.

### 6.2 The impact of government regulators on the amount of fines imposed on drug production enterprises and third-party testing agencies

Set $F_p = \{0.0, 5.0, 10.0\}$, $F_t = \{0.0, 5.0, 10.0\}$, the evolution process and results of the main strategies of the four-party game are shown in Fig 7.

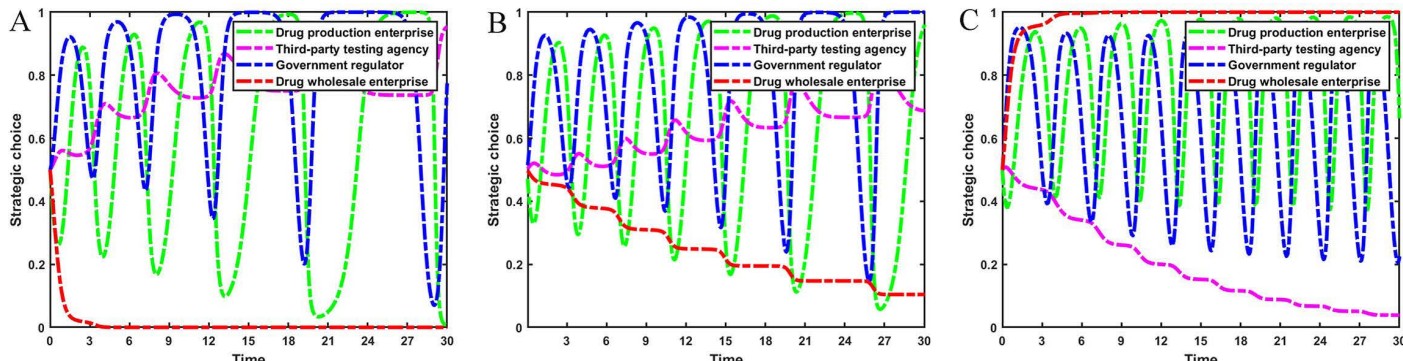

**Fig 6. The impact of fines and incentives on drug wholesale enterprises.** Fig 6 is the simulation diagram that shows the influence of the reward amount brought by drug wholesale enterprises reporting and the fine amount brought by drug wholesale enterprises not reporting on the strategic choices of drug production enterprises, third-party testing agencies, government regulators, and drug wholesale enterprises. **(A)** When the situation $F_w = 0.0$, $M_w = 0.0$. **(B)** When the situation $F_w = 3.0$, $M_w = 3.0$. **(C)** When the situation $F_w = 7.0$, $M_w = 7.0$.

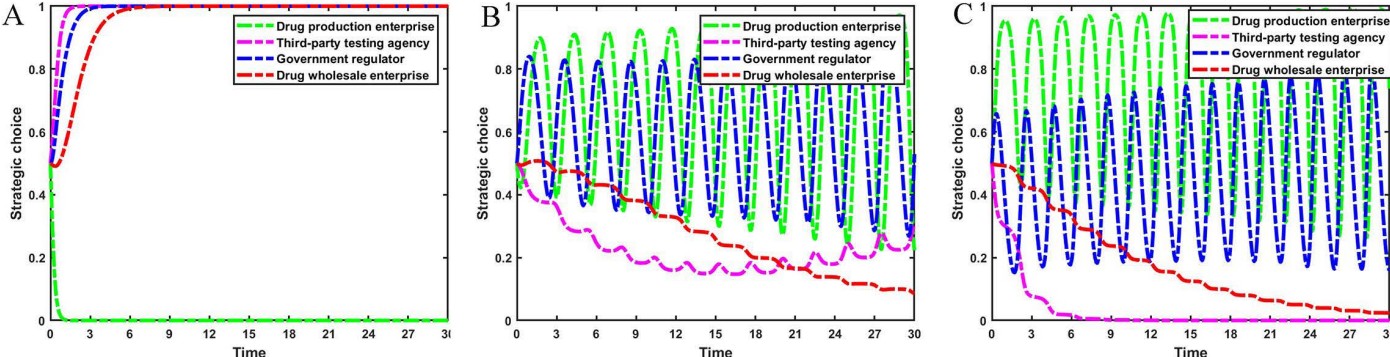

**Fig 7. The impact of fines on drug production enterprises and third-party testing agencies.** Fig 7 is the simulation diagram that shows the influence of the fine amount brought by drug production enterprises dishonest management and third-party testing agencies accept rent-seeking on the strategic choices of drug production enterprises, third-party testing agencies, government regulators, and drug wholesale enterprises. **(A)** When the situation $F_p = 0.0$, $F_t = 0.0$. **(B)** When the situation $F_p = 5.0$, $F_t = 5.0$. **(C)** When the situation $F_p = 10.0$, $F_t = 10.0$.

As can be seen from Fig 7, the increase in fines imposed by government regulators on drug production enterprises and third-party testing agencies will help drug production enterprises choose an integrity management strategy. When government regulators do not punish dishonest business practices and rent-seeking behaviors, drug production enterprises operate dishonestly and third-party testing agencies accept rent-seeking, resulting in low-quality drugs flowing into the market. Government regulators have increased strict supervision in order to standardize the order of the pharmaceutical market, and drug wholesale enterprises have chosen to report rent-seeking behaviors. As the fines imposed by government regulators for dishonest business practices and accepting rent-seeking behavior increase, drug production enterprises' strategies change more frequently and there is a higher probability of their choosing integrity management strategy. Third-party testing agencies accept rent-seeking behavior with a lower probability. At this time, the pharmaceutical market is relatively stable, drug wholesale enterprises will reduce the probability of reporting, and the rate of strict supervision by government regulators will tend to decline.

## 6.3 Impact of reporting by drug wholesale enterprises

Set $z = \{0.0, 0.5, 1.0\}$, the evolution process and results of the main strategies of the four-party game are shown in Fig 8.

Fig 8 shows that the reporting behavior of drug wholesale enterprises can help curb the rent-seeking phenomenon between drug production enterprises and third-party testing agencies. When drug wholesale enterprises fail to report low-quality drugs, speculation about dishonest management by drug production enterprises continues to occur, and third-party testing agencies are intent on rent-seeking. At this time, the strategies of government regulators are adjusted accordingly. As the probability of reporting by drug wholesale enterprises increases, the speculation of dishonest management by drug wholesale enterprises decreases. Third-party testing agencies gradually change their strategies to reject rent-seeking. The frequency of government regulators' policy changes gradually decreases and they are more inclined to strict supervision strategies.

## 6.4 The impact of strict supervision by government regulators

Set $g = \{0.0, 0.5, 1.0\}$, the evolution process and results of the main strategies of the four-party game are shown in Fig 9.

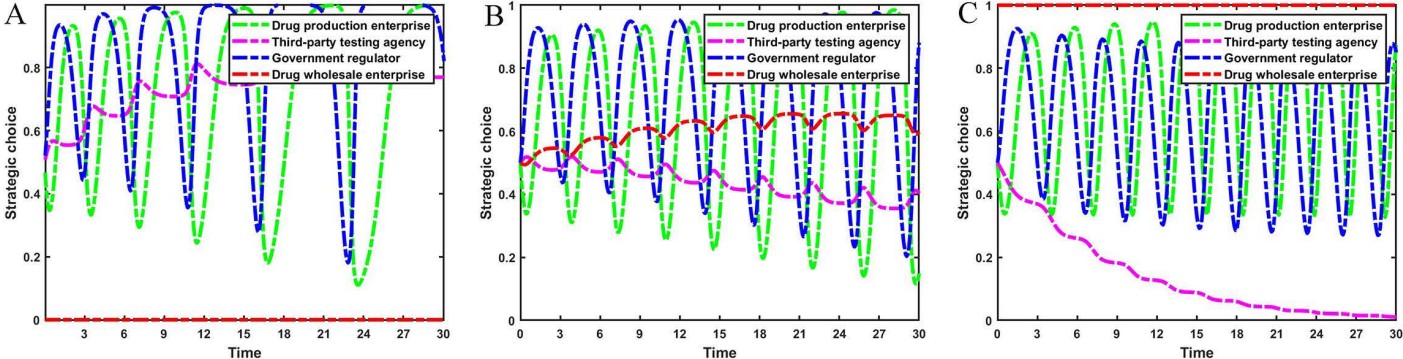

**Fig 8. The impact of reporting by drug wholesale enterprises.** Fig 8 is the simulation diagram that shows the influence of the probability of reporting by drug wholesale enterprises on the strategic choices of drug production enterprises, third-party testing agencies, and government regulators. **(A)** When the situation $z = 0.0$. **(B)** When the situation $z = 0.5$. **(C)** When the situation $z = 1.0$.

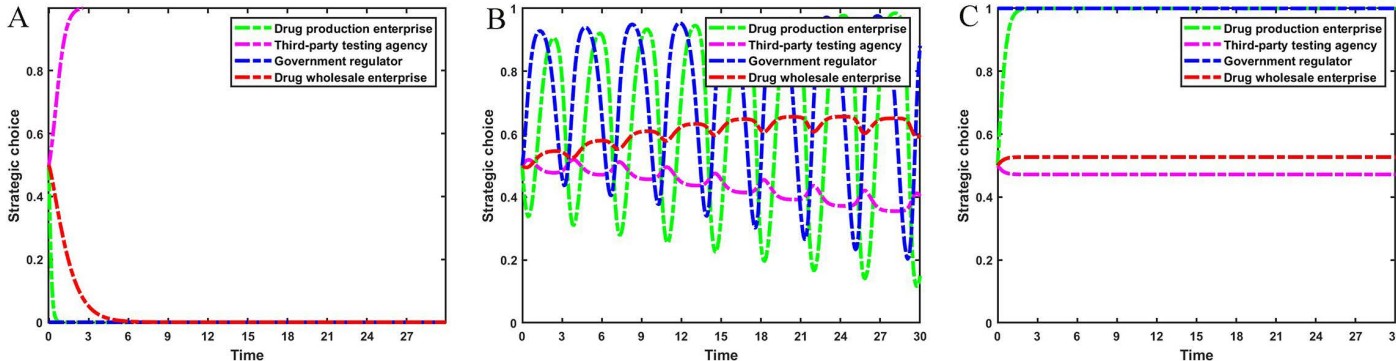

**Fig 9. The impact of strict supervision by government regulators.** Fig 9 is the simulation diagram that shows the influence of the probability of strict supervision by government regulators on the strategic choices of drug production enterprises, third-party testing agencies, and drug wholesale enterprises. **(A)** When the situation $g = 0.0$. **(B)** When the situation $g = 0.5$. **(C)** When the situation $g = 1.0$.

Fig 9 shows that strengthening strict supervision by government regulators will also help curb the occurrence of rent-seeking between drug production enterprises and third-party testing agencies. When the stabilization strategy of the government regulators is to loosen supervision, drug production enterprises operate dishonestly and third-party testing agencies accept rent-seeking, and drug wholesale enterprises wholesale low-quality drugs but do not report them. This is the least ideal situation at this time. As government regulators increase the probability of strict supervision, the strategies of drug production enterprises become unstable, and the strategic choices of third-party testing agencies are adjusted accordingly. Reporting phenomenon by drug wholesale enterprises continue to occur. The stable strategy of government regulators is that strict supervision is the most ideal state for the drug market. Drug production enterprises operate with integrity and produce high-quality drugs, third-party testing agencies pass high-quality drugs through testing, and drug wholesale enterprises wholesale and sell high-quality drugs.

## 6.5 Impact of cost

Suppose a drug wholesale enterprise discovers low-quality drugs and reports them. Set $B_t = \{2.9, 4.9, 6.9\}$, $C_1 - C_2 = \{13.0, 8.0, 3.0\}$ and $C_r = \{4.0, 5.0, 6.0\}$. The strategic evolution

process and results of the three-party game involving drug production enterprises, third-party testing agencies, and government regulators are illustrated in Fig 10.

Fig 10 shows that when the additional costs of strict supervision by government regulators are high, and the rent-seeking costs of drug production enterprises and the reporting costs of drug wholesale enterprises are low, there is a strategic stability point (0, 0, 0) in the replication dynamic system. At this time, government regulators chose to loosen supervision in order to reduce costs, drug production enterprises operated dishonestly, and third-party testing agencies refused to seek rent. As the additional costs required for strict supervision by government regulators decrease, and the rent-seeking costs of drug production enterprises and the reporting costs of drug wholesale enterprises increase, the replication dynamic system is in an unstable state. When the additional costs of strict supervision are further reduced, and the rent-seeking costs and reporting costs are further increased, the replication dynamic system will be stable at the strategy combination (0, 1, 1). At this time, third-party testing agencies accept rent-seeking, drug production enterprises operate dishonestly, and government regulators strictly supervise. Thus, the loose supervision by government regulators and the acceptance of rent-seeking by third-party testing agencies hinder the honest operation of drug production enterprises. Government regulators should improve regulatory efficiency, reduce the cost of strict supervision, and improve the probability of honest operation by improving reporting channels, reducing reporting costs, and increasing the rent-seeking costs of drug production enterprises.

## 7 Discussions

This study constructs an evolutionary game model for the integrity supervision of pharmaceutical enterprises, analyzing the influence of various key factors on strategy evolution and proposing targeted decision-making suggestions for all parties. While exploring the theoretical effects of the reward and punishment mechanism, we also recognize the complexity and challenges in implementing it under the existing regulatory framework.

Firstly, drug production enterprises, as the cornerstone of market integrity, their behaviors are directly related to public health and industry reputation. They should actively choose integrity-based operation and establish a complete internal monitoring and compliance system to ensure product quality and safety. Especially when facing third-party testing agencies, they need to adhere to transparent cooperation and resolutely resist rent-seeking behaviors.

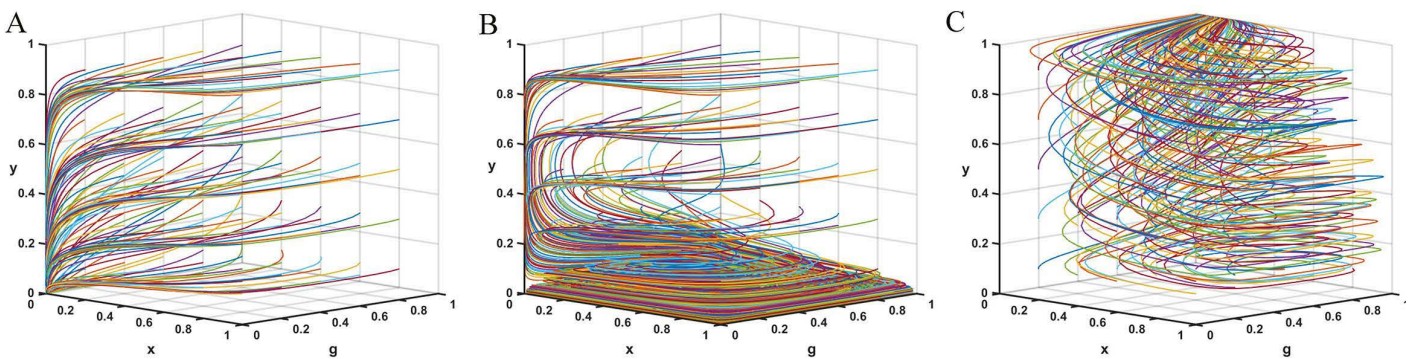

**Fig 10. Impact of cost.** Fig 10 is the simulation diagram that shows the influence of various costs on the strategic choices of drug production enterprises, third-party testing agencies, and government regulators. **(A)** When the situation $B_t = 2.9$ , $C_1 - C_2 = 13.0$ , $C_r = 4.0$ . **(B)** When the situation $B_t = 4.9$ , $C_1 - C_2 = 8.0$ , $C_r = 5.0$ . **(C)** When the situation $B_t = 6.9$ , $C_1 - C_2 = 3.0$ , $C_r = 6.0$ .

Under the guidance of the government's reward and punishment mechanism, enterprises should strengthen communication with regulatory departments and participate in integrity evaluations to enhance their corporate images.

Secondly, the credibility of third-party testing agencies is crucial to the healthy development of the pharmaceutical market. Strengthening internal management and ensuring the fairness and transparency of testing processes are the keys to enhancing their international competitiveness. Refusing rent-seeking behaviors not only consolidates their own market positions but also significantly enhances the willingness of enterprises to operate with integrity. Testing agencies should regard fair testing as their mission. By actively accepting supervision, improving testing levels and service qualities, they can win market trust, avoid short-term interests harming long - term reputations, and promote the healthy growth of the global pharmaceutical industry.

In addition, the strictness and fairness of government regulators are the last line of defense for maintaining market order and public health. By optimizing the reward and punishment mechanism, they can not only encourage integrity-based operation but also effectively curb rent-seeking and dishonesty. However, although theoretical analysis shows that increasing the punishment intensity has a significant effect on suppressing rent-seeking behaviors, under the existing regulatory framework, the implementation of the reward and punishment mechanism faces multiple challenges. The limited regulatory resources restrict regulatory agencies from comprehensively monitoring illegal behaviors, especially in highly specialized and information-asymmetric fields such as the pharmaceutical industry. Excessive punishment may inhibit the innovation motivation of enterprises, while too-light punishment cannot form an effective deterrence. Therefore, when designing the reward and punishment mechanism, the government needs to balance the intensity of punishment within the legal framework, and at the same time consider industry characteristics, enterprise scales, and the nature of illegal behaviors to ensure the fairness and rationality of the mechanism. Specifically, the government should start from two aspects. Firstly, clarify and increase the economic punishment intensity for rent-seeking behaviors of pharmaceutical enterprises to make it exceed a certain threshold. By significantly increasing the amount of fines to exceed the potential benefits brought by rent-seeking behaviors, the illegal cost of enterprises can be effectively increased. For serious dishonest behaviors, the economic punishment should be sufficient to weaken the competitiveness of enterprises in the market and may even lead to enterprise bankruptcy liquidation to form a strong deterrent effect. Secondly, the government should consider implementing criminal punishments. In key areas involving public health and safety, the pursuit of criminal liability is crucial. For behaviors of serious rent-seeking or causing major damage consequences, the government can pursue the criminal liability of relevant responsible persons in accordance with the law, implement punitive detention, deprive of business qualifications and other measures, thereby enhancing the deterrence of the law. In addition, in order to effectively curb rent-seeking, the blacklist system should be strengthened, a public supervision and reporting mechanism should be established, and multi-department joint law enforcement and other measures should be considered. Although these policy tools will increase regulatory costs in the short term, in the long run, by improving regulatory effects and reducing illegal behaviors, the overall regulatory cost can be reduced, and at the same time, market confidence and stability can be enhanced, thereby reducing the costs of supervision and law enforcement.

Finally, drug wholesale enterprises, as the bridge connecting production and consumption, are crucial in maintaining market order through the reporting mechanism. Enterprises should establish and improve internal reporting and feedback mechanisms to ensure that problem products are reported quickly. The government should provide convenient reporting channels

for wholesale enterprises, protect the rights and interests of whistleblowers, and give appropriate rewards to stimulate their enthusiasm for participating in supervision.

In conclusion, the government, enterprises, and third-party testing agencies need to flexibly adjust strategies according to their respective national conditions to jointly promote the construction of the integrity supervision system in the pharmaceutical industry and promote the healthy development of the global pharmaceutical market and the protection of public health. However, it is worth noting that the model construction and analysis in this study are mainly based on the game theory framework, assuming that all participants make decisions based on bounded rational behavior. This assumption simplifies the complexity of the real world to a certain extent. Therefore, although the model can reveal the effect of the reward and punishment mechanism under ideal conditions, when applied to actual scenarios, the uncertainty brought by irrational behaviors and the possibility of deviation from predicted results need to be considered. To make up for this limitation, future research can try to introduce the perspective of behavioral economics, incorporate irrational behaviors into model consideration, in order to more comprehensively reflect the decision-making process in the real world and thus improve the prediction accuracy and application value of the model.

## 8 Conclusions

Promoting the modernization of integrity supervision in the pharmaceutical industry requires the participation of all stakeholders. We should enhance the integrity awareness of drug production enterprises, increase policy support, reduce their costs of operating with integrity, and encourage them to consciously choose to operate with integrity. Additionally, we should urge third-party testing agencies to reject rent-seeking, increase the speculative costs of third-party testing agencies accepting rent-seeking by improving the professionalism of testing personnel and expanding media disclosure, and encourage them to assist the government in strictly supervising pharmaceutical enterprises. Fully leveraging the indirect regulatory role of drug wholesale enterprises and encouraging them to actively report untrustworthy behaviors of drug production enterprises are also essential. The introduction of reward and punishment mechanisms and the participation of third-party testing agencies are of great significance for promoting the diversification of subjects and methods of integrity supervision in the pharmaceutical industry and improving supervision efficiency.

Considering the strategy choices of drug production enterprises, third-party testing agencies, government regulators, and drug wholesale enterprises, this paper constructs a four-party evolutionary game model. This paper studies the role of the government's reward and punishment mechanism in the integrity supervision of pharmaceutical enterprises. We realize that the reputation mechanism of social media, with its unique reputation punishment and reward functions, also plays an indispensable role in the integrity supervision of pharmaceutical enterprises. Therefore, future research should deeply explore the quantitative influence of the social media reputation mechanism, including the specific correlation between public feedback on social media and corporate behaviors, and how to further enhance the integrity awareness of pharmaceutical enterprises by optimizing social media supervision policies, so as to construct a more complete and effective integrity supervision system. Through empirical analysis of the role of the social media reputation mechanism in the integrity supervision of pharmaceutical enterprises, data support can be provided for policy-makers, guiding the government, enterprises, and third-party agencies on how to use social media to improve supervision efficiency in practical operations, and jointly create a more integrity-based, transparent, and healthy pharmaceutical market environment.

## Supporting information

**S1 File. Supporting information.**
(DOCX)

## Author contributions

**Formal analysis:** Lilong Zhu.

**Software:** Yanhua Chen.

**Supervision:** Lilong Zhu.

**Writing – original draft:** Yanhua Chen.

**Writing – review & editing:** Lilong Zhu.

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
