## [Decision Letter · Decision Letter 0]

15 Oct 2024

PONE-D-24-28268Pharmaceutical Enterprises Integrity Supervision Strategy When Considering Rent-seeking Behavior and Government Reward and Punishment MechanismPLOS ONE

Dear Dr. Zhu,

Thank you for submitting your manuscript to PLOS ONE. After careful consideration, we feel that it has merit but does not fully meet PLOS ONE’s publication criteria as it currently stands. Therefore, we invite you to submit a revised version of the manuscript that addresses the points raised during the review process.

The manuscript has been evaluated by four reviewers, and their comments are available below.

The reviewers have raised a number of concerns that need attention. They request additional elaboration in the introduction and discussion with additional details of implications for this work, as well as further clarity in the results.

We look forward to receiving your revised manuscript.

Kind regards,

Avanti Dey, PhD

Staff Editor

PLOS ONE

Journal Requirements:

Additional Editor Comments (if provided):

Reviewers' comments:

Reviewer's Responses to Questions

**Comments to the Author**

1. Is the manuscript technically sound, and do the data support the conclusions?

Reviewer #1: Yes

Reviewer #2: Yes

Reviewer #3: Yes

Reviewer #4: Yes

2. Has the statistical analysis been performed appropriately and rigorously? 

Reviewer #1: Yes

Reviewer #2: Yes

Reviewer #3: N/A

Reviewer #4: No

3. Have the authors made all data underlying the findings in their manuscript fully available?

Reviewer #1: Yes

Reviewer #2: Yes

Reviewer #3: Yes

Reviewer #4: Yes

4. Is the manuscript presented in an intelligible fashion and written in standard English?

Reviewer #1: Yes

Reviewer #2: Yes

Reviewer #3: Yes

Reviewer #4: Yes

5. Review Comments to the Author

Reviewer #1: Q1�Please introduce the definition of rent-seeking behavior and its practical consequences in the introduction section.

Q2�In the abstract, there is one of the findings " Reducing the costs of stringent government supervision and increasing the speculative costs of rent-seeking for thirdparty testing agencies help prevent dishonest practices among drug production enterprises". (1) Why the impact of regulatory costs and rent-seeking costs are discussed on the dishonest practices among drug production enterprises? (2) The guidance on the decision-making of all parties should be added in the discussion section.

Q3: Please explain why evolutionary game theory is used to solve the problem in this paper.

Q4: There are some grammar errors.

Q5: The content of the manuscript is too long. The proofs can be appropriately simplified.

Reviewer #2: I would like to thank the esteemed editor for inviting me to review this manuscript. I have carefully reviewed this article and the following issues need to be addressed before publication. My specific comments are as follows:

1. The introductory section provides an overview of pharmaceutical corporate integrity and its importance, but could be further enhanced with a critical analysis of the existing literature, clearly identifying the shortcomings of existing research and the innovations of this study.

2. subsection 2.1 refers to pharmaceutical corporate integrity issues, but does not elaborate on how these issues specifically affect public health and industry development.

3. Subsection 2.2 discusses government regulation of integrity and suggests adding some analyses on cases of regulatory failures to enhance the persuasiveness of the argument.

4. The model assumptions (H1-H6) provide the basis for model construction, but there is a need to ensure that these assumptions are well grounded and supported by data in the real world.

5. In the model construction section, it is recommended that more details be provided, such as the selection of variables and the method of determining parameters.

6. the section provides mathematical analyses of the stability of the strategies of different participants. It is recommended that visual explanations of these mathematical models be added to aid understanding by non-specialist readers. In addition, the authors should include in the analysis a sensitivity analysis for different parameter variations. It is recommended to refer to the following references.-Tripartite evolutionary game analysis and simulation research on zero-carbon production supervision of marine ranching against a carbon-neutral background�A tripartite evolutionary game study of low-carbon innovation system from the perspective of dynamic subsidies and taxes Research on the tripartite evolutionary game of public participation in the facility location of hazardous materials logistics from the perspective of NIMBY events

7. The simulation analyses were partially conducted using Matlab 2022b, but no detailed description of the code or the models was provided. To increase transparency and reproducibility, it is recommended that the simulation code or at least key code segments be provided.

8 The graphs (Figure 6 to Figure 10) provide a visual representation of the simulation results, but there is a need to ensure that each graph has a detailed legend and explanation.

9. The discussion section provides some policy recommendations, but it is recommended that the applicability and feasibility of these recommendations in different countries and regions be explored in more depth.

10. The Recommendations Discussion section includes a discussion of the limitations of the model and how these limitations may affect the findings and recommendations.

11.The conclusion section succinctly summarises the key findings of the study. It is recommended that the authors consider adding specific recommendations for future research directions and how these findings can be applied in practice.

Overall, this paper provides valuable insights and modelling analysis in the area of integrity regulation of pharmaceutical companies. However, in order to improve the quality and impact of the paper, the authors need to make detailed revisions and additions to the issues raised above.

Reviewer #3: This work should be published after addressing the following points.

• Could you add parameters in Figure 1? It is difficult to understand the relationship among parameters from the current manuscript.

• The information in Section 3.1 is hard to understand. A table related to the hypotheses would be helpful.

• Is the model itself novel? Please clarify it.

• Which assumption is the most critical for your results? Also, which parameter is the most sensitive for your results? Please clarify them quantitatively.

• How do you manage uncertainties of your calculation results? There should be many uncertainties in your model.

• What is the key message of Figure 10? The review believes the figure is the most important result in the manuscript, so please clarify the message.

Reviewer #4: Review report

This paper offers a valuable contribution to the understanding of pharmaceutical enterprise integrity, presenting a robust theoretical model backed by quantitative analysis. However, there are several areas that require improvement, particularly concerning practical applicability, empirical validation, and a more focused discussion of the complex realities of pharmaceutical regulation.

Major/Minor Comments

• The paper contributes to pharmaceutical integrity through a strong theoretical model but needs improvement in practical applicability, empirical validation, and real-world complexity.

• The discussion on enforcing theoretical findings like increased penalties for rent-seeking is insufficient, with little exploration of challenges in applying the reward and punishment mechanisms in existing regulatory frameworks.

• The model assumes rational behavior, overlooking factors like political pressure or hidden conflicts of interest that could affect stakeholder decisions.

• Key external factors such as globalization, technological advancements, and international regulatory variations are missing, limiting the study's scope.

• Theoretical outcomes lack real-world validation through case studies or empirical data, weakening the model's practical credibility.

• Policy recommendations are too broad and vague, particularly regarding reducing supervision costs and increasing penalties for rent-seeking behavior. More detailed, actionable policy tools and cost-benefit analyses are required.

• The model would benefit from real-world scenario discussions, providing clearer guidance for policymakers.

• Extending the model to include non-rational behavior, international influences, or corruption would offer a more realistic representation of pharmaceutical regulation.

• Case studies or empirical data from various regulatory environments would help validate the model’s predictions and identify necessary refinements.

• Recommendations on penalties for rent-seeking should explore specific forms (financial, criminal) and enforcement methods.

• Ethical implications of rent-seeking and the role of corporate governance should be discussed to offer a more comprehensive approach to integrity.

6. PLOS authors have the option to publish the peer review history of their article (what does this mean? ). If published, this will include your full peer review and any attached files.

**Do you want your identity to be public for this peer review?** For information about this choice, including consent withdrawal, please see our Privacy Policy .

Reviewer #1: No

Reviewer #2: No

Reviewer #3: **Yes: ** Yusuke Hayashi

Reviewer #4: No

---

## [Author Response · Author response to Decision Letter 0]

30 Oct 2024

Responses to Editor and Reviewers

Dear Editor Professor Avanti Dey and Reviewers,

Thank you very much for giving us an opportunity to submit a revised version of the manuscript entitled “Pharmaceutical Enterprises Integrity Supervision Strategy When Considering Rent-seeking Behavior and Government Reward and Punishment Mechanism.” The manuscript is a revised submission (Manuscript ID: PONE-D-24-28268) with new page numbers in the text, some grammar and spelling errors have also been corrected. The revised paper contains 36 pages, 10 figures and 4 tables.

We also thank the reviewers for their critical reading and valuable feedback on the manuscript. These comments provide direction for our further research. According to the reviewer's comments, relevant modifications have been made to the original manuscript, with the main changes marked in different colors. We mark the comments of the first reviewer in purple, the comments of the second reviewer in blue, the comments of the third reviewer in green, and the suggestions of the fourth reviewer in red. We also responded to each reviewer's comments listed below one by one and clearly indicated the location of the revisions. In addition, after careful consideration, we have made further improvements to the paper without changing its structure and general content, and have marked it in bold blue font.

Responds to the editor ’s comments:

Reviewer #1

Comment #1:

Please introduce the definition of rent-seeking behavior and its practical consequences in the introduction section.

Response: Thank you very much for your suggestion. As you suggested, we should introduce the definition of rent-seeking behavior and its practical consequences in the introduction part. We have added to the first paragraph of the introduction (the first paragraph on the second page). Added as: "The integrity of pharmaceutical enterprises is an embodiment of national governance capabilities and an important guarantee for the healthy development of the economy and society. The lack of integrity in the pharmaceutical field seriously affects the normal operation of the medical industry, especially when rent-seeking behaviors are involved. Rent-seeking behavior refers to the fact that enterprises or individuals obtain economic benefits through non-market means, such as bribery or interest transfer, rather than achieving profit growth through improving efficiency or innovation. In the pharmaceutical industry, drug production enterprises influence the test results of third-party testing agencies through rent-seeking behavior, which not only distorts normal medical behaviors and reduces the quality of medical services, but also may allow unqualified drugs to enter the market, seriously threatening the life safety of patients. From an ethical perspective, rent-seeking behavior places economic interests above patients' health and is a betrayal of professional spirit and social responsibility. At the same time, rent-seeking behavior leads to unfair allocation of medical resources, exacerbates excessive medical treatment and drug price increases, increases the economic burden on patients, and ultimately hinders the healthy and sustainable development of the pharmaceutical industry. Against this background, it is urgent to strengthen corporate governance to ensure that corporate behaviors comply with ethical norms and legal regulations, thereby reshaping industry trust and promoting a virtuous cycle in the pharmaceutical industry."

Comment #2:

In the abstract, there is one of the findings "Reducing the costs of stringent government supervision and increasing the speculative costs of rent-seeking for third-party testing agencies help prevent dishonest practices among drug production enterprises". (1) Why the impact of regulatory costs and rent-seeking costs are discussed on the dishonest practices among drug production enterprises? (2) The guidance on the decision-making of all parties should be added in the discussion section.

Response: (1) Thank you for your in-depth attention and professional suggestions on our research. We explore the influence of supervision costs and rent-seeking costs on the dishonest behaviors of drug production enterprises. The core purpose is to reveal and quantify how these costs play a role in the integrity supervision of pharmaceutical enterprises. The level of supervision costs directly affects the supervision intensity and efficiency of government regulators, and then determines the binding force on pharmaceutical enterprises. When the supervision costs are too high, it may weaken the supervision ability of government regulators and provide opportunities for the dishonest behaviors of drug production enterprises; conversely, reasonable and appropriate supervision costs can effectively deter illegal behaviors and promote the integrity-based operation of enterprises. Meanwhile, rent-seeking costs, that is, the risks and costs that enterprises need to bear in order to evade supervision, are directly related to whether they choose to avoid supervision through illegal means such as bribery. Higher rent-seeking costs can significantly suppress the rent-seeking motives of enterprises, forcing them to tend to abide by the law with integrity; otherwise, it may intensify unfair competition in the market and affect the health of the industry. Therefore, in-depth analysis of the specific impacts of supervision costs and rent-seeking costs on drug production enterprises not only provides theoretical support for constructing an efficient supervision system but also has important guiding significance for actual policy formulation and implementation, which is helpful to form a good situation of strict government supervision, enterprise self-discipline and social co-governance, and safeguard the healthy development of the pharmaceutical industry and the health rights and interests of the public.

(2) Thank you very much for your suggestion. As you suggested, we have revised the discussion part and provided more specific suggestions for each entity (on page 29). We added, "This paper constructs an evolutionary game model for the integrity supervision of pharmaceutical enterprises, analyzes the influence of various key factors on strategy evolution, and proposes targeted decision-making suggestions for all parties. When deeply exploring the theoretical effects of the reward and punishment mechanism, we also realize the complexity and challenges of its implementation under the existing regulatory framework.

Firstly, drug production enterprises, as the cornerstone of market integrity, their behaviors are directly related to public health and industry reputation. They should actively choose integrity-based operation and establish a complete internal monitoring and compliance system to ensure product quality and safety. Especially when facing third-party testing agencies, they need to adhere to transparent cooperation and resolutely resist rent-seeking behaviors. Under the guidance of the government's reward and punishment mechanism, enterprises should strengthen communication with regulatory departments and participate in integrity evaluations to enhance their corporate images.

Secondly, the credibility of third - party testing agencies is crucial to the healthy development of the pharmaceutical market. Strengthening internal management and ensuring the fairness and transparency of testing processes are the keys to enhancing their international competitiveness. Refusing rent-seeking behaviors not only consolidates their own market positions but also significantly enhances the willingness of enterprises to operate with integrity. Testing agencies should regard fair testing as their mission. By actively accepting supervision, improving testing levels and service qualities, they can win market trust, avoid short-term interests harming long - term reputations, and promote the healthy growth of the global pharmaceutical industry.

In addition, the strictness and fairness of government regulators are the last line of defense for maintaining market order and public health. By optimizing the reward and punishment mechanism, they can not only encourage integrity-based operation but also effectively curb rent-seeking and dishonesty. However, although theoretical analysis shows that increasing the punishment intensity has a significant effect on suppressing rent-seeking behaviors, under the existing regulatory framework, the implementation of the reward and punishment mechanism faces multiple challenges. The limited regulatory resources restrict regulatory agencies from comprehensively monitoring illegal behaviors, especially in highly specialized and information-asymmetric fields such as the pharmaceutical industry. Excessive punishment may inhibit the innovation motivation of enterprises, while too-light punishment cannot form an effective deterrence. Therefore, when designing the reward and punishment mechanism, the government needs to balance the intensity of punishment within the legal framework, and at the same time consider industry characteristics, enterprise scales, and the nature of illegal behaviors to ensure the fairness and rationality of the mechanism. Specifically, the government should start from two aspects. Firstly, clarify and increase the economic punishment intensity for rent-seeking behaviors of pharmaceutical enterprises to make it exceed a certain threshold. By significantly increasing the amount of fines to exceed the potential benefits brought by rent-seeking behaviors, the illegal cost of enterprises can be effectively increased. For serious dishonest behaviors, the economic punishment should be sufficient to weaken the competitiveness of enterprises in the market and may even lead to enterprise bankruptcy liquidation to form a strong deterrent effect. Secondly, the government should consider implementing criminal punishments. In key areas involving public health and safety, the pursuit of criminal liability is crucial. For behaviors of serious rent-seeking or causing major damage consequences, the government can pursue the criminal liability of relevant responsible persons in accordance with the law, implement punitive detention, deprive of business qualifications and other measures, thereby enhancing the deterrence of the law. In addition, in order to effectively curb rent-seeking, the blacklist system should be strengthened, a public supervision and reporting mechanism should be established, and multi-department joint law enforcement and other measures should be considered. Although these policy tools will increase regulatory costs in the short term, in the long run, by improving regulatory effects and reducing illegal behaviors, the overall regulatory cost can be reduced, and at the same time, market confidence and stability can be enhanced, thereby reducing the costs of supervision and law enforcement.

Finally, drug wholesale enterprises, as the bridge connecting production and consumption, are crucial in maintaining market order through the reporting mechanism. Enterprises should establish and improve internal reporting and feedback mechanisms to ensure that problem products are reported quickly. The government should provide convenient reporting channels for wholesale enterprises, protect the rights and interests of whistleblowers, and give appropriate rewards to stimulate their enthusiasm for participating in supervision.

In conclusion, the government, enterprises, and third-party testing agencies need to flexibly adjust strategies according to their respective national conditions to jointly promote the construction of the integrity supervision system in the pharmaceutical industry and promote the healthy development of the global pharmaceutical market and the protection of public health. However, it is worth noting that the model construction and analysis in this study are mainly based on the game theory framework, assuming that all participants make decisions based on bounded rational behavior. This assumption simplifies the complexity of the real world to a certain extent. Therefore, although the model can reveal the effect of the reward and punishment mechanism under ideal conditions, when applied to actual scenarios, the uncertainty brought by irrational behaviors and the possibility of deviation from predicted results need to be considered. To make up for this limitation, future research can try to introduce the perspective of behavioral economics, incorporate irrational behaviors into model consideration, in order to more comprehensively reflect the decision-making process in the real world and thus improve the prediction accuracy and application value of the model."

Comment #3:

Please explain why evolutionary game theory is used to solve the problem in this paper.

Response: Thank you for your attention and guidance on our research. We choose game theory as an analytical tool mainly because the issue of integrity supervision of pharmaceutical enterprises involves interactions and strategy selections among multiple entities, which is exactly the core research area of game theory. In the context of integrity supervision in the pharmaceutical industry, the relationships among drug production enterprises, third-party testing agencies, government regulators, and drug wholesale enterprises are complex. Each party makes decisions based on its own interests, and these decisions influence each other, forming a dynamic strategic game. By constructing an evolutionary game model, we can systematically analyze the strategy selection processes of each entity, understand the influence of the reward and punishment mechanism on strategy evolution, and then explore the optimal strategy combination, which provides a theoretical basis for improving supervision efficiency and promoting integrity-based operation. In addition, the game theory analysis also helps us quantify the influence of each decision variable on the strategy selections of different entities. Through simulation analysis, the effectiveness of the theoretical model is verified, providing specific guidance for policy formulation and practical application. Therefore, we have made a supplement in the Problem Description part of 3.1 (the last paragraph on page six). The supplementary content is: "Evolutionary game theory, as a powerful analytical tool, has the core advantage of being able to depict and predict the dynamic strategic interactions among multiple agents. It is especially suitable for research scenarios involving complex decision-making environments such as integrity supervision of pharmaceutical enterprises. By using evolutionary game theory, we can deeply analyze the decision-making process of each participating party when facing the reward and punishment mechanism and reveal the mechanism of action of the reward and punishment mechanism on strategy evolution. Therefore, evolutionary game theory has strong applicability and effectiveness in solving the problem of integrity supervision of pharmaceutical enterprises. "

Comment #4:

There are some grammar errors.

Response: Sincerely thank you for the valuable comments you put forward on our paper. The grammar mistakes you pointed out are indeed the areas that we need to attach importance to and correct. We deeply understand the importance of accurate expression for the quality of academic papers. We have now carried out a detailed grammar check and correction on the full text to improve the readability and professionalism of the paper.

Comment #5:

The content of the manuscript is too long. The proofs can be appropriately simplified.

Response: Thank you very much for your valuable suggestions. We understand that the conciseness of the paper content is crucial for clearly conveying the research focus. We have followed your advice to appropriately simplify the proof process, ensuring that the paper content is more concise while still maintaining the integrity and rigor of the argument.

Reviewer #2

Comment #1:

The introductory section provides an overview of pharmaceutical corporate integrity and its importance, but could be further enhanced with a critical analysis of the existing literature, clearly identifying the shortcomings of existing research and the innovations of this study.

Respons

---

## [Decision Letter · Decision Letter 1]

31 Jan 2025

PONE-D-24-28268R1Pharmaceutical Enterprises Integrity Supervision Strategy When Considering Rent-seeking Behavior and Government Reward and Punishment MechanismPLOS ONE

Dear Dr. Zhu,

Thank you for submitting your manuscript to PLOS ONE. After careful consideration, we feel that it has merit but does not fully meet PLOS ONE’s publication criteria as it currently stands. Therefore, we invite you to submit a revised version of the manuscript that addresses the points raised during the review process.

The manuscript has been assessed by the Editorial team and we have some remaining revision requests:

1. Please work to improve the quality of the writing throughout your manuscript. We recommend asking a colleague who is proficient in written English to assist you; alternatively, you could enlist the help of a professional copyediting service.

2. We note that you have published a paper with a similar methodology and focus in the journal Complexity (https://onlinelibrary.wiley.com/doi/10.1155/2021/5865299). Given the similarity in some of the hypotheses between this paper and your current submission, it needs to be made clear how these two studies differ in terms of the research questions being asked and the current study framed within the context of the literature. Please clarify this in either the covering letter accompanying the revised version or within the text itself.

Could you please revise the manuscript to carefully address the concerns raised?

We look forward to receiving your revised manuscript.

Kind regards,

Helen Howard

Staff Editor

PLOS ONE

Journal Requirements:

Reviewers' comments:

Reviewer's Responses to Questions

**Comments to the Author**

1. If the authors have adequately addressed your comments raised in a previous round of review and you feel that this manuscript is now acceptable for publication, you may indicate that here to bypass the “Comments to the Author” section, enter your conflict of interest statement in the “Confidential to Editor” section, and submit your "Accept" recommendation.

Reviewer #2: All comments have been addressed

Reviewer #3: All comments have been addressed

Reviewer #4: All comments have been addressed

2. Is the manuscript technically sound, and do the data support the conclusions?

Reviewer #2: Yes

Reviewer #3: Yes

Reviewer #4: Yes

3. Has the statistical analysis been performed appropriately and rigorously? 

Reviewer #2: Yes

Reviewer #3: Yes

Reviewer #4: Yes

4. Have the authors made all data underlying the findings in their manuscript fully available?

Reviewer #2: Yes

Reviewer #3: Yes

Reviewer #4: Yes

5. Is the manuscript presented in an intelligible fashion and written in standard English?

Reviewer #2: Yes

Reviewer #3: Yes

Reviewer #4: Yes

6. Review Comments to the Author

Reviewer #2: Well done. The authors have addressed my concerns, and I recommend publishing this version.

This paper incorporates rent-seeking behavior and introduces a reward and punishment mechanism to construct an evolutionary game model involving drug

production enterprises, third-party testing agencies, government regulators, and drug wholesale enterprises. By solving for the stable equilibrium points of each participant's strategic choices and analyzing the stability of strategy combinations using Lyapunov's first method, the study employs Matlab 2022b for simulation analysis to verify the impact of various decision variables on the strategic choices of different entities.This paper enriches the theoretical foundation of pharmaceutical integrity supervision and offers pertinent countermeasures and recommendations.

Reviewer #3: (No Response)

Reviewer #4: The authors have addressed all the concerns and comments. I personally congratulate the authors of the manuscript on their joint efforts.

7. PLOS authors have the option to publish the peer review history of their article (what does this mean? ). If published, this will include your full peer review and any attached files.

**Do you want your identity to be public for this peer review?** For information about this choice, including consent withdrawal, please see our Privacy Policy .

Reviewer #2: No

Reviewer #3: **Yes: ** Yusuke Hayashi

Reviewer #4: **Yes: ** Saad Salman

---

## [Author Response · Author response to Decision Letter 1]

13 Feb 2025

Dear Editor Professor Helen Howard and Reviewers,

Thank you very much for giving us an opportunity to submit a revised version of the manuscript entitled “Pharmaceutical Enterprises Integrity Supervision Strategy When Considering Rent-seeking Behavior and Government Reward and Punishment Mechanism.” The manuscript is a revised submission (Manuscript ID: PONE-D-24-28268) with new page numbers in the text, some grammar and spelling errors have also been corrected. The revised paper contains 36 pages, 10 figures and 4 tables.

We also thank the reviewers for their critical reading and the valuable feedback from the peer reviewers. These comments provide directions for our further research. According to the reviewers' opinions, the original manuscript was modified accordingly, and the main modifications were marked with different colors. We will mark the modification of the first opinion in blue, and the modification of the second opinion in red, and reply to each opinion listed below one by one, and clearly point out the location of the revision.

Responds to the editor ’s comments:

Comment #1:

Please work to improve the quality of the writing throughout your manuscript. We recommend asking a colleague who is proficient in written English to assist you; alternatively, you could enlist the help of a professional copyediting service.

Response: Thank you very much for your suggestion. We are well aware of the importance of accurate expression to the quality of academic papers. We have carried out a detailed grammar check and revision of the full text to improve the readability and professionalism of the paper, and have made some modifications to improve the writing quality of the manuscript and strive to make the original manuscript more excellent.

Comment #2:

We note that you have published a paper with a similar methodology and focus in the journal Complexity (https://onlinelibrary.wiley.com/doi/10.1155/2021/5865299). Given the similarity in some of the hypotheses between this paper and your current submission, it needs to be made clear how these two studies differ in terms of the research questions being asked and the current study framed within the context of the literature. Please clarify this in either the covering letter accompanying the revised version or within the text itself.

Response: Thank you for your careful review and valuable comments. In the materials currently submitted, we have carefully considered your concerns. The research of Zhang and zhu[11] mainly focused on the collaborative supervision between local governments, pharmaceutical enterprises and third-party testing agencies. They focused on how to improve the integrity management level of pharmaceutical enterprises through the participation of third-party testing agencies under the government reward and punishment mechanism. Their research focuses on how to promote the honest operation of pharmaceutical enterprises and reduce the flow of unqualified drugs into the market through the synergy of local governments and third-party testing agencies. On the basis of Zhang and Zhu, the current research further introduces the role of pharmaceutical wholesale enterprises, and constructs a four parties evolutionary game model including pharmaceutical enterprises, third-party testing agencies, government regulators and pharmaceutical wholesale enterprises. Our research questions not only include how to promote the honest operation of pharmaceutical enterprises through the government reward and punishment mechanism, but also pay special attention to the impact of drug wholesale enterprises' reporting behavior on the strategic choice of pharmaceutical enterprises and third-party testing agencies. In addition, we also discussed how to improve the efficiency of government regulation and promote the integrity level of the entire drug supply chain by optimizing the reward and punishment mechanism.

We have added relevant explanations on page 5 of the original text. The specific content is "The research by Zhang and Zhu [11] has provided valuable insights into the co-regulation of pharmaceutical enterprises. However, their study has certain limitations. Firstly, their research only considered the relationships among local governments, drug enterprises, and third-party testing agencies, while ignoring the important role of drug wholesale enterprises in the supervision of drug quality. Secondly, their model assumptions are relatively simple and do not fully take into account the whistleblowing behavior of drug wholesale enterprises and its impact on the strategic choices of all parties. Compared with their study, this research introduces drug wholesale enterprises as a new participant and constructs a more complex four-party evolutionary game model, which can more comprehensively reflect the actual game relationship in the pharmaceutical supply chain. " We hope these clarifications can address your concerns. Thank you again for your time and attention.

We have tried our best to improve the manuscript and made some substantial changes and necessary deletions according to the editors’ and reviewers’ comments. We earnestly appreciate the editors’ and reviewers’ professional work and hope that the corrections will make our manuscript suitable for publication in PLOS ONE. We are looking forward to receiving comments from reviewers in the future. If you have any questions, please do not hesitate to contact me at the address below.

Once again, thank you very much for your valuable comments and suggestions.

Best wishes.

Sincerely,

Lilong Zhu, Ph.D. and Professor. Postdoctoral research in Shandong University. Research scholar in College of business, University of Illinois at Urbana-Champaign, USA. Ph.D. degree was granted in management science and engineering from Tongji University and Postdoctoral research in Shandong University. His research interests are supply chain management and quality management.

E-mail: zhulilong2008@126.com Tel: +86-13853193366

College of Business, Shandong Normal University, School of Management, Shandong University, Ji’nan 250014, Shandong, China.

---

## [Editor Report · Decision Letter 2]

27 Feb 2025

Pharmaceutical Enterprises Integrity Supervision Strategy When Considering Rent-seeking Behavior and Government Reward and Punishment Mechanism

PONE-D-24-28268R2

Dear Dr. Zhu,

We’re pleased to inform you that your manuscript has been judged scientifically suitable for publication and will be formally accepted for publication once it meets all outstanding technical requirements.

Kind regards,

Helen Howard

Staff Editor

PLOS ONE
---

## [Editor Report · Acceptance letter]

PONE-D-24-28268R2

PLOS ONE

Dear Dr. Zhu,

I'm pleased to inform you that your manuscript has been deemed suitable for publication in PLOS ONE. Congratulations! Your manuscript is now being handed over to our production team.

Kind regards,

on behalf of

Dr Helen Howard

Staff Editor

PLOS ONE